# POLCAM: instant molecular orientation microscopy for the life sciences

**Ezra Bruggeman** [1,2], **Oumeng Zhang** [3], **Lisa-Maria Needham**[1,4], **Markus Körbel** [1], **Sam Daly** [1], **Matthew Cheetham**[1], **Ruby Peters**[5], **Tingting Wu** [3], **Andrey S. Klymchenko** [6], **Simon J. Davis**[7], **Ewa K. Paluch** [5], **David Klenerman** [1], **Matthew D. Lew** [3], **Kevin O'Holleran** [8] & **Steven F. Lee** [1,2] ✉

Current methods for single-molecule orientation localization microscopy (SMOLM) require optical setups and algorithms that can be prohibitively slow and complex, limiting widespread adoption for biological applications. We present POLCAM, a simplified SMOLM method based on polarized detection using a polarization camera, which can be easily implemented on any wide-field fluorescence microscope. To make polarization cameras compatible with single-molecule detection, we developed theory to minimize field-of-view errors, used simulations to optimize experimental design and developed a fast algorithm based on Stokes parameter estimation that can operate over 1,000-fold faster than the state of the art, enabling near-instant determination of molecular anisotropy. To aid in the adoption of POLCAM, we developed open-source image analysis software and a website detailing hardware installation and software use. To illustrate the potential of POLCAM in the life sciences, we applied our method to study α-synuclein fibrils, the actin cytoskeleton of mammalian cells, fibroblast-like cells and the plasma membrane of live human T cells.

Single-molecule localization microscopy (SMLM)[1–4] is a super-resolution microscopy technique that is widely used in biology to study cellular structures below the diffraction limit[5–7]. Single-molecule orientation localization microscopy (SMOLM) is a multidimensional variant of SMLM in which, in addition to the precise spatial position, the orientation of individual fluorescent molecules is also measured. The ability to measure the orientation of single molecules provides information about how molecules organize, orient, rotate and wobble in their environment, which is of key relevance across biological systems[8–13]. The widespread use of SMOLM by the biological imaging community has thus far been limited by the need for complex experimental setups and often computationally expensive image analysis. Additionally, this lack of accessibility has also slowed down the necessary development of a wider range of labeling protocols that are appropriate for SMOLM: labeling methods in which the orientation of the fluorescent probe is relatively fixed and rotationally restricted with respect to its target[14–17].

Fluorescent molecules are not isotropic point sources, that is, they do not emit light equally in all directions. Fundamentally, fluorescent molecules emit like oscillating electric dipoles: the intensity *I* of the emitted fluorescence depends on the relative observation direction and follows the relationship $I \propto \sin^2(\eta)$, where $\eta$ is the angle between the observation direction and the orientation of the emission dipole moment of the molecule[18]. In conventional SMLM experiments, this anisotropic emission is typically not noticeable because, with common

[1]Yusuf Hamied Department of Chemistry, University of Cambridge, Cambridge, UK. [2]Aligning Science Across Parkinson's (ASAP) Collaborative Research Network, Chevy Chase, MD, USA. [3]Department of Electrical and Systems Engineering, Washington University in St. Louis, St. Louis, MO, USA. [4]Department of Chemistry, University of Wisconsin–Madison, Madison, WI, USA. [5]Department of Physiology, Development and Neuroscience, University of Cambridge, Cambridge, UK. [6]Laboratoire de Biophotonique et Pharmacologie, Université de Strasbourg, Strasbourg, France. [7]MRC Weatherall Institute of Molecular Medicine, University of Oxford, Oxford, UK. [8]Cambridge Advanced Imaging Centre, University of Cambridge, Cambridge, UK. ✉e-mail: sl591@cam.ac.uk

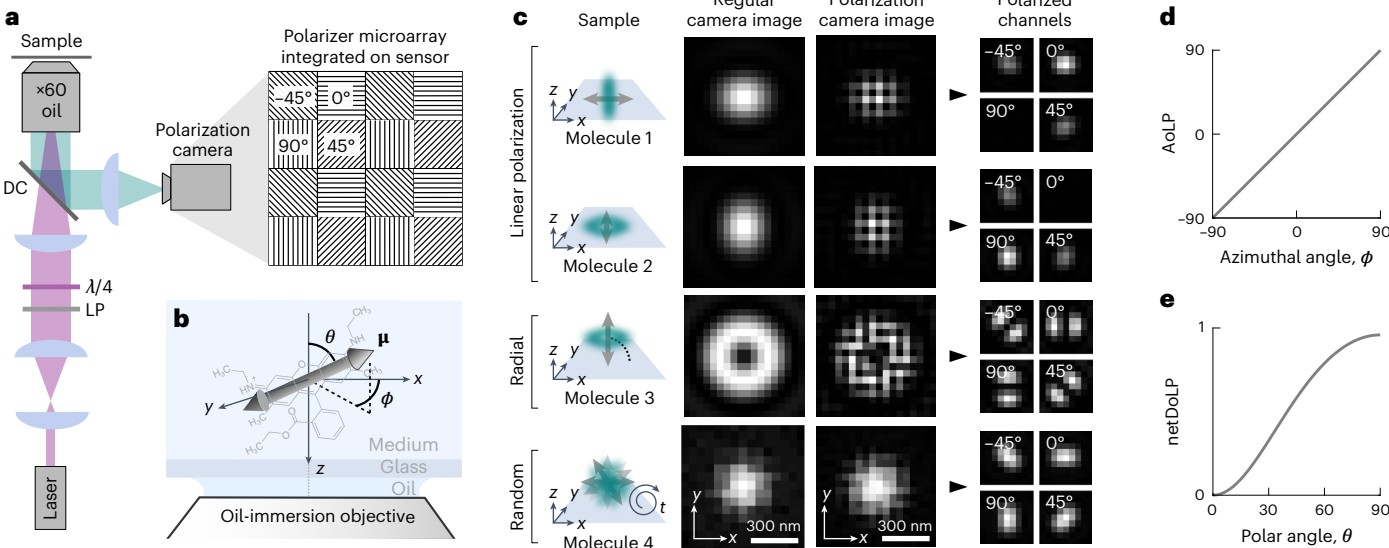

**Fig. 1 | Single-molecule imaging using a polarization camera. a**, Schematic of the optical setup that includes the polarization camera and a schematic representation of a small region of the four-directional micropolarizer array (transmission axis at 0°, 45°, 90° or −45°) integrated into the sensor. LP, linear polarizer; λ/4, quarter-wave plate; DC, dichroic. **b**, Definition of the in-plane angle $\phi$ and the out-of-plane angle $\theta$ that specify the orientation of the emission dipole moment (arrow) of a molecule (structure of rhodamine 6G depicted). **c**, Simulated examples of four single fluorescent molecules. From top to bottom, a molecule aligned with the $x$ axis (first row), the $y$ axis (second row), the optical axis ($z$ axis) (third row) and a rapidly rotating molecule (fourth row). For each

example, the following is shown: the image plane recorded with a regular monochrome camera and the image plane recorded with a polarization camera in its raw format and in a format where the pixels have been rearranged to form four images that are each made up only of pixels that are covered by a micropolarizer with the same transmission axis orientation. **d**, Relationship between the average AoLP and the in-plane angle $\phi$ of the dipole moment. **e**, Relationship between the netDoLP and the out-of-plane angle $\theta$ of the dipole moment of a rotationally immobilized molecule for a 1.4-NA oil-immersion objective and no refractive index mismatch between the sample and the immersion medium.

SMLM labeling protocols, fluorescent molecules are free to rapidly wobble and rotate with respect to their target (for example, due to long linker chains), resulting in an orientation-averaged image[19]. However, when a labeling method is used that restricts the rotational freedom of the fluorescent molecules with respect to their target, the anisotropy in the emitted fluorescence allows for the measurement of molecular orientation[20]. Different approaches have been used to achieve molecular orientation imaging. Some methods are based on active modulation of the polarization of the excitation light[21–23], but the majority of methods are based on modification of the detection path of the microscope. The image of a single molecule can be fitted using a dipole-spread function (DSF) that includes the position of the molecule ($x, y$ or $x, y, z$), the orientation of the emission dipole moment ($\phi, \theta$)[24,25] and often a rotational mobility parameter[26]. As the intensity distribution of a standard DSF does not contain notable information about a molecule's orientation, the DSF can be engineered to increase the orientation information content. A simple example is imaging slightly out of focus to exaggerate the DSF shape[27–29]. More advanced DSF engineering can be performed using a spatial light modulator[19,30–33], a special optic[34–36] or pupil splitting[37]. A drawback of DSF engineering is that the optical setups required are highly complex (with the exception of the vortex DSF[34]) and are sensitive to optical aberrations (with the exception of pupil splitting[37]), as the orientation estimation algorithms rely on simulated DSF models, which can necessitate performing spatially (in) variant phase retrieval to match the DSF model to the experimental DSF[34]. Additionally, fitting a five- or six-dimensional DSF model is computationally expensive, making data analysis prohibitively slow.

An alternative method is splitting the emission into multiple polarized channels that form separate images on the same camera[10,38–44] or multiple detectors[45] or using continuous image displacement using a rotating calcite crystal[46]. The advantage of polarized detection-based methods is that the orientation estimation can be performed using simple intensity measurements, and does not necessarily require the

fitting of a complex DSF model. As a result, the data analysis is fast and easily compatible with high-throughput data collection. The simplest polarized detection setup splits the emission into two orthogonal polarized channels using a polarizing beam splitter[38,39]. This method suffers from some degeneracies because both an isotropic emitter (for example, a freely rotating molecule or fluorescent bead) and an immobilized molecule with an emission dipole moment oriented at 45° in between the transmission axis of the two channels or parallel to the optical axis all give rise to equal intensities measured in both channels[8,47]. Splitting the emission into three or more polarized channels or cameras breaks this degeneracy[15,48,49] but substantially complicates the experimental setup[39,41–44].

Here, we present a new experimentally simplified SMOLM method called POLCAM that uses a polarization camera for four-channel polarized detection. Polarization cameras have become popular in the field of computer vision as they provide single-shot multi-channel polarized measurements[50–53]. The pixels of the sensor of a polarization camera are covered by small linear polarizers with transmission axes typically oriented at 0°, 45°, 90° and −45° in a 2 × 2-pixel mosaic pattern that is repeated over the entire sensor (Fig. 1a). As polarizers are integrated into the camera chip, no additional polarization optics are required. Polarization cameras have reached quantum efficiencies and noise levels that in theory are compatible with single-molecule detection, but the ease of use of polarization cameras does come at a cost in the form of instantaneous field-of-view (IFOV) errors near object edges[54–56]. To make POLCAM robust to IFOV errors, we used vectorial diffraction simulations of single dipole emitters to optimize microscope design and developed a Stokes parameter estimation-based reconstruction algorithm and a DSF-fitting algorithm.

We validated and characterized our method using samples with known structures: single fluorophores immobilized on coverglass and in polymer and lipid bilayer-coated glass beads labeled with membrane dyes. We next performed SMOLM on amyloid fibrils in vitro and on the actin

network of fixed mammalian cells. To demonstrate that a polarization camera can also be used for conventional polarized detection microscopy, we imaged the actin network of COS-7 cells and the membrane of live human T cells interacting with an antibody-coated coverglass.

Our approach can be easily implemented by changing the regular camera on any single-molecule fluorescence microscope to a polarization camera. The camera used in this work is supported by the popular image acquisition software µManager[57], and we provide image analysis software in the form of MATLAB applications for single-molecule and diffraction-limited image analysis and real-time image processing and rendering during acquisition and a napari[58] plugin for processing multidimensional diffraction-limited polarization camera image datasets. We envisage that the combination of ease of use, ease of implementation, low cost, improved speed and open-source software will make POLCAM an accessible and powerful tool for the study of molecular orientation across diverse biological applications.

## Results

### Measuring molecular orientation using polarized detection

When a fluorescent molecule has one dominant emission dipole moment $\mu$, as is the case for many common fluorescent molecules, its emission resembles the far field emitted by an oscillating electric dipole[18,20] (Fig. 1b,c and Supplementary Notes 1 and 3). When its emission dipole moment is oriented parallel to the sample plane, the electric field in the back focal plane of the objective will mainly be linearly polarized along the direction of the emission dipole moment (Fig. 1c, first and second row). If the emission dipole moment is oriented parallel to the optical axis, the electric field in the back focal plane of the objective will be radially polarized[59,60] (Fig. 1c, third row). When a molecule is rapidly rotating, it can appear unpolarized (Fig. 1c, bottom row). As the tube lens in conventional wide-field fluorescence microscopy has a low numerical aperture (NA), the described polarization is mostly conserved in the image plane[61]. As a result, the angle of the axis of maximum polarization determines the in-plane orientation $\phi$ of the emission dipole moment, and the degree of net linear polarization is related to the out-of-plane orientation $\theta$ (Fig. 1d,e).

Conventionally, polarizing beam splitters are used to split the detected fluorescence into multiple polarized image channels. The number of photons that are detected from a single molecule in the different channels will depend on the three-dimensional (3D) orientation and rotational mobility of the molecule. These measured intensities can be used to estimate the angles $\phi$ and $\theta$ using analytically derived equations. Equations for the case of four polarized detection channels (0°, 45°, 90° and −45°) were derived by John T. Fourkas[60]. Here, we rewrote these expressions in terms of Stokes parameters (Supplementary Notes 4 and 5):

$$\phi = \frac{1}{2}\tan^{-1}\left(\frac{S_2}{S_1}\right) = \text{AoLP} \qquad (1)$$

$$\theta = \sin^{-1}\left(\sqrt{\frac{A \times \text{netDoLP}}{C - B \times \text{netDoLP}}}\right), \qquad (2)$$

where $A$, $B$ and $C$ are constants that are a function of the half-maximum collection angle of the objective $\alpha$ (Supplementary Notes 4 and 5) and the net degree of linear polarization (netDoLP) given by

$$\text{netDoLP} = \sqrt{\frac{\langle S_1 \rangle^2 + \langle S_2 \rangle^2}{\langle S_0 \rangle^2}}, \qquad (3)$$

where the brackets $\langle ... \rangle$ refer to averaging over a small region of interest around a single molecule (Supplementary Note 7) and $S_0$, $S_1$ and $S_2$ are the first three Stokes parameters:

$$S_0 = (I_0 + I_{45} + I_{90} + I_{-45})/2 \qquad (4a)$$

$$S_1 = I_0 - I_{90} \qquad (4b)$$

$$S_2 = I_{45} - I_{-45}, \qquad (4c)$$

where $I_0$, $I_{45}$, $I_{90}$ and $I_{-45}$ refer to the measured intensities in the four polarized channels. The full derivation of equations (1) and (2) can be found in Supplementary Notes 4 and 5, and example simulated images are in Supplementary Figs. 3–19.

We note that equation (1) is simply the expression for the angle of linear polarization (AoLP)[62] and that equation (2) depends only on the netDoLP and the objective used. This is in line with our intuition (Fig. 1c). Equations (1) and (2) are plotted in Fig. 1d,e for a 1.4-NA oil-immersion objective. We note that estimation of $\phi$ using equation (1) is very robust but that estimation of $\theta$ using equation (2) on the other hand is only possible under strict conditions (perfect rotational immobilization, high signal-to-noise ratio and no large refractive index mismatch) and is more reliably performed with DSF fitting as will be discussed in more detail in a later section.

As a proxy for rotational mobility, we use the average degree of linear polarization (avgDoLP), which we define as a local average of the degree of linear polarization (DoLP):

$$\text{avgDoLP} = \sum_{i=1}^{m} \text{DoLP}_i = \sum_{i=1}^{m} \sqrt{\frac{S_{1,i}^2 + S_{2,i}^2}{S_{0,i}^2}} \qquad (5)$$

where $m$ is the number of pixels in the region of interest around the molecule. If a DSF-fitting algorithm is used instead ('Improving accuracy by considering the DSF shape'), a rotational mobility parameter $\gamma$ can be estimated that inversely relates to the size of a cone in which the molecule has rotational freedom[26]: $\gamma$ is 1 for perfect immobilization and 0 for complete rotational freedom. The exact mathematical relation between avgDoLP and $\gamma$ is complex, as avgDoLP is also influenced by the signal-to-noise ratio, the out-of-plane angle and the refractive index of the sample medium. The relationship between avgDoLP and rotational mobility is numerically explored in Supplementary Fig. 27, showing that it is monotonic under all conditions and can therefore be used qualitatively but with care. For details, we refer to the Methods and Supplementary Note 7.

### Overcoming IFOV errors

In conventional polarization-sensitive fluorescence imaging, where polarizing elements are placed in the optical path, a traditional camera sensor captures the full-intensity distribution everywhere in the image plane. However, similar to the operation of conventional color image sensors[63], the polarization camera measures the intensity of each polarization channel in a subset of the image (one in four pixels), and the full-intensity distributions have to be recovered through interpolation[54–56]. This recovery can be performed accurately provided that the pixel size is small enough. If this is not the case, any measurements taken from the recovered channels will exhibit what is known as IFOV errors[54–56]. In standard applications of polarization cameras (for example, quality control in the manufacturing industry, removal of reflections in images[64]), IFOV errors can be ignored or avoided, as the pixel size can be much smaller than variations in neighboring pixels and artifacts mostly appear near the edges of objects. However, when imaging single molecules, the opposite is true, as the image of a single emitter varies substantially over each pixel and the limited photon budget prevents the use of a small pixel size.

We determined the optimal pixel size for the estimation of molecular orientation using vectorial diffraction simulations (Supplementary Note 6). We define the optimal pixel size as the largest pixel size that

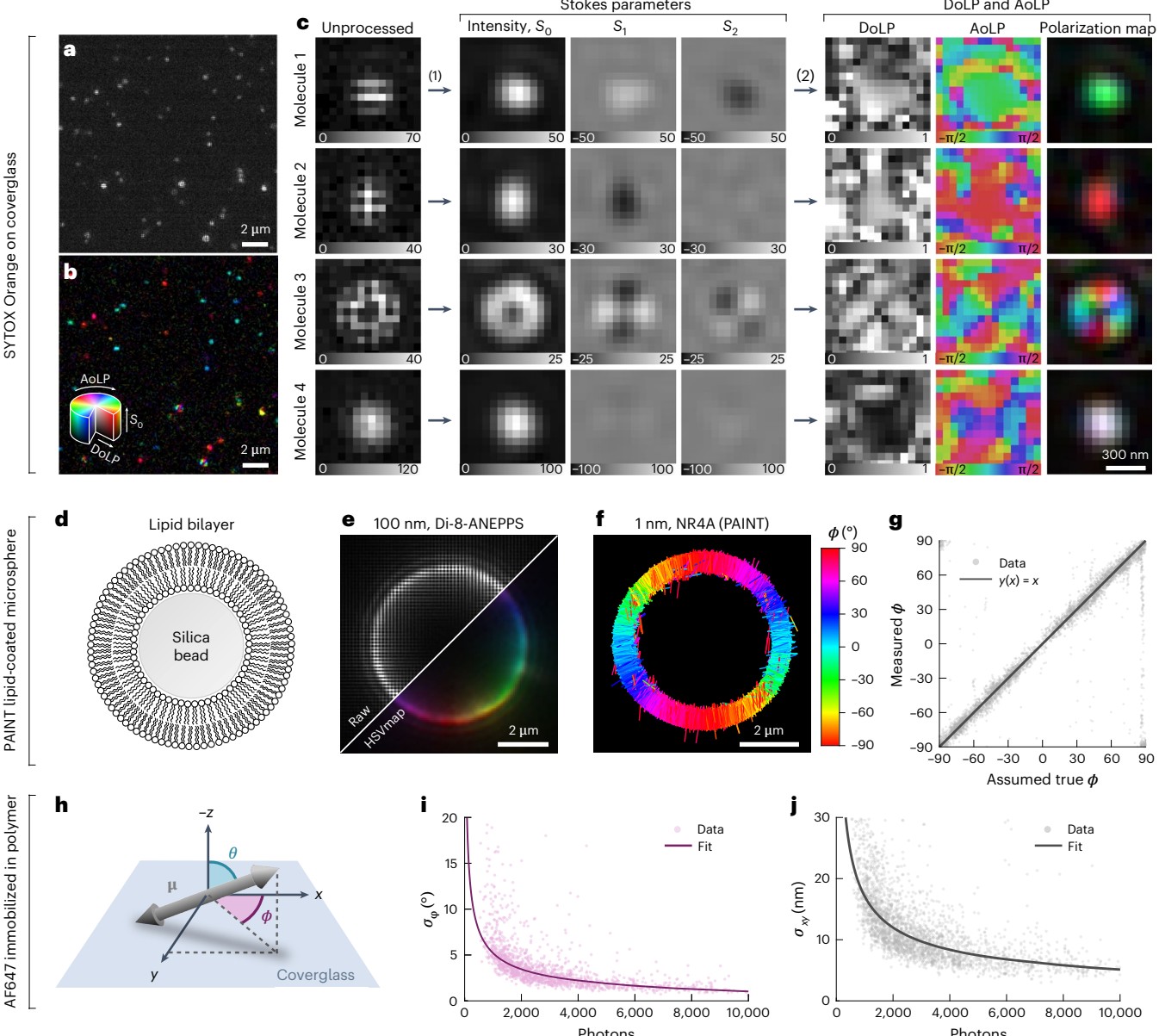

**Fig. 2 | Single-molecule detection, experimental bias and precision. a,** An unprocessed polarization camera image of SYTOX Orange molecules dispersed on a coverglass in PBS. **b,** The same image as in **a** but processed to reveal polarization information using a polarization color map that combines the AoLP, the DoLP and the intensity ($S_0$) in HSV (hue, saturation, value) color space (hue, AoLP; saturation, DoLP; value, $S_0$). **c,** Examples of four SYTOX Orange molecules with their emission dipole moment parallel to the sample plane (molecules 1 and 2), parallel to the optical axis (molecule 3) and rapidly rotating (molecule 4). For each, the unprocessed image, estimated Stokes parameter images ($S_0$, $S_1$ and $S_2$), the AoLP, the DoLP and polarization color map images are shown. **d,** Illustration of a silica microsphere (5 μm in diameter) coated using a lipid bilayer (DPPC with 40% cholesterol). **e,** Diffraction-limited image of a cross-section at a z plane in the middle

of a lipid bilayer-coated silica microsphere labeled using the membrane dye Di-8-ANEPPS. **f,** POLCAM SMOLM reconstruction of a cross-section of a lipid bilayer-coated silica microsphere acquired through PAINT with Nile red. Each localization is drawn as a rod with a direction indicating the estimated in-plane angle $\phi$. **g,** An experimental bias curve for the estimation of $\phi$ generated using a PAINT dataset such as the one shown in **f. h,** Illustration of the angles specifying the orientation of the emission dipole moment. **i,j,** Experimental precision from repeated localization and orientation estimation on AF647 immobilized in polyvinyl alcohol (PVA). The precision is the measured standard deviation on repeated measurement of the position ($x, y$) (**j**) and the in-plane angle $\phi$ (**i**) of the same molecule. Photon numbers are averages. Measurements between $n = 12$ and $n = 40$ are used to calculate the standard deviation. A power law was fitted to the data.

still allows for accurate recovery of the four polarized channels from a single polarization camera image. To assess whether accurate recovery is possible, we used an approach described by Tyo et al.[55] that checks for overlap between the contributions of different Stokes parameters in the Fourier transform of the unprocessed polarization camera image (Supplementary Note 6.1 and Supplementary Figs. 20–25). If the contributions do not overlap, the recovery is assumed to be accurate.

Using this method, we find that, for our setup (1.4-NA oil-immersion objective, wavelength of 650 nm, sample in an aqueous

medium), optimal sampling is achieved at a pixel size of ~60 nm × 60 nm (Supplementary Fig. 20). Practically, a pixel size of 57.5 nm × 57.5 nm was achieved using a ×60 magnification objective and the standard polarization camera pixel size of 3.45 μm × 3.45 μm (Supplementary Fig. 30). The calculated ideal pixel size as a function of wavelength, objective NA and sample medium can be found in Supplementary Fig. 20.

Next, we compared the performance of different algorithms[54–56] for Stokes parameter estimation and channel interpolation on

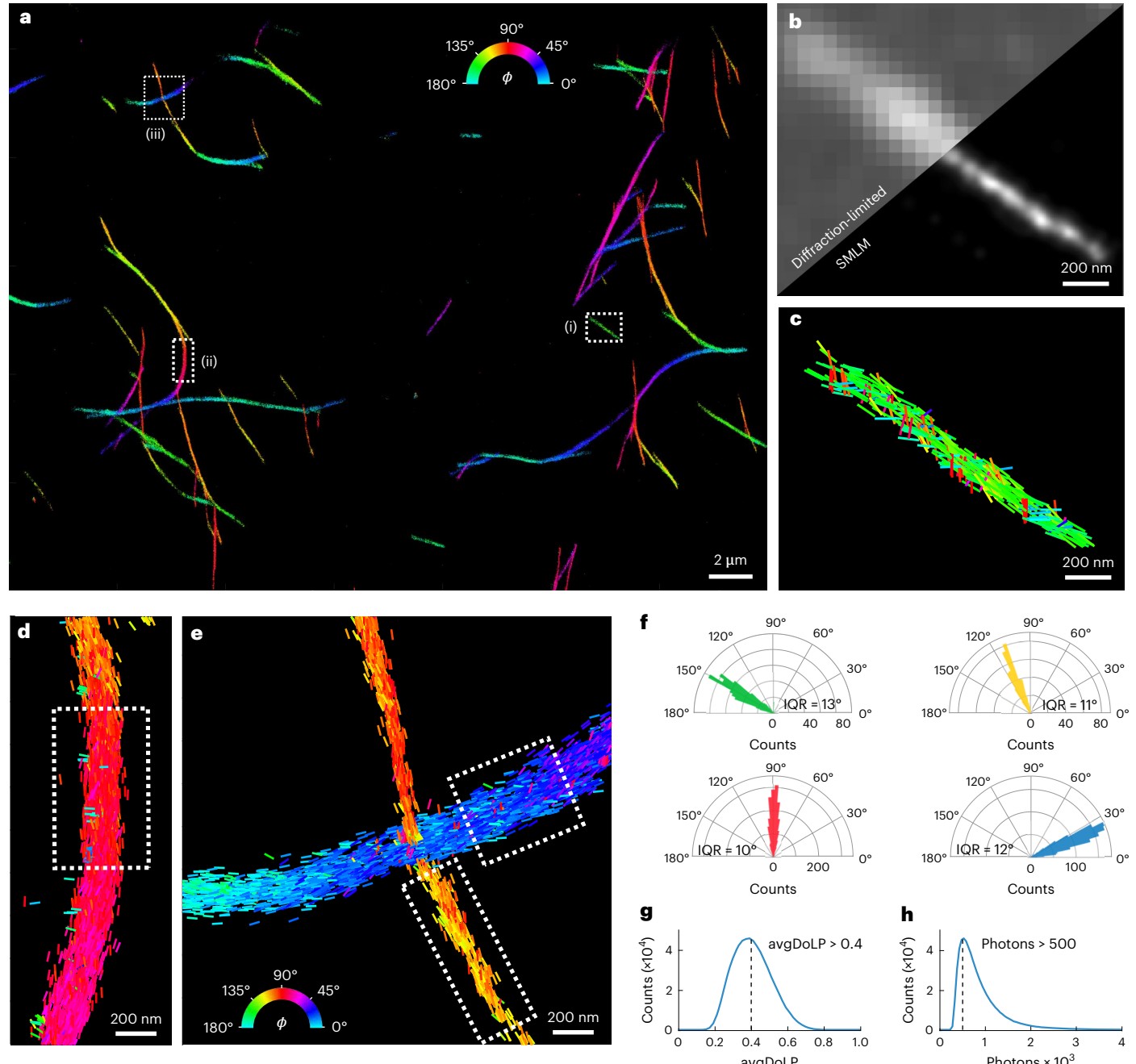

**Fig. 3 | TAB-PAINT imaging of α-synuclein fibrils in vitro. a**, A POLCAM SMOLM reconstruction of α-synuclein fibrils, color coded by the in-plane angle $\phi$ of the emission dipole moment of the Nile red molecules. **b**, Diffraction-limited image and SMLM reconstruction of inset (i) from **a**. **c**–**e**, Detail of insets (i) (**c**), (ii) (**d**) and (iii) (**e**) in **a**, where individual localizations are drawn as rods. The orientation and color of the rods indicate the measured $\phi$. **f**, Polar histograms of the fibrils shown in **c** (green), and segments are indicated by the dotted boxes drawn in **d** (red) and **f** (yellow and blue). The interquartile range (IQR) of the distributions is displayed below the respective histograms. **g**, Histogram of the avgDoLP of the data shown in **a**, including the avgDoLP threshold used to exclude localizations with high rotational mobility when displaying $\phi$-color-coded reconstructions. **h**, Histogram of the number of detected photons per molecule per frame for the dataset shown in **a**. The minimum number of detected photons (500 photons) is also indicated.

simulated polarization camera images of immobilized single molecules. We found that a Fourier-based approach[55] and cubic spline interpolation performed the best (Supplementary Note 6.2 and Supplementary Figs. 21–25). Fig. 2a shows an experimental polarization camera image of semi-immobilized SYTOX Orange molecules on a coverglass. Fig. 2b shows the results of the Fourier-based interpolation. Fig. 2c shows examples of the emission of single molecules taken from this dataset: two in-plane-oriented molecules (molecules 1 and 2), one molecule oriented out of the plane (molecule 3) and a rapidly rotating

molecule (molecule 4). An example SYTOX Orange dataset is included in Supplementary Dataset 1.

### Experimental accuracy and precision

To measure the experimental accuracy of orientation estimation, silica microspheres with a diameter of 5 µm were coated with a lipid bilayer (dipalmitoylphosphatidylcholine (DPPC)) containing 40% cholesterol; Fig 2d) as described in refs. 33,37 and labeled using different membrane dyes. Polarization-resolved diffraction-limited images were

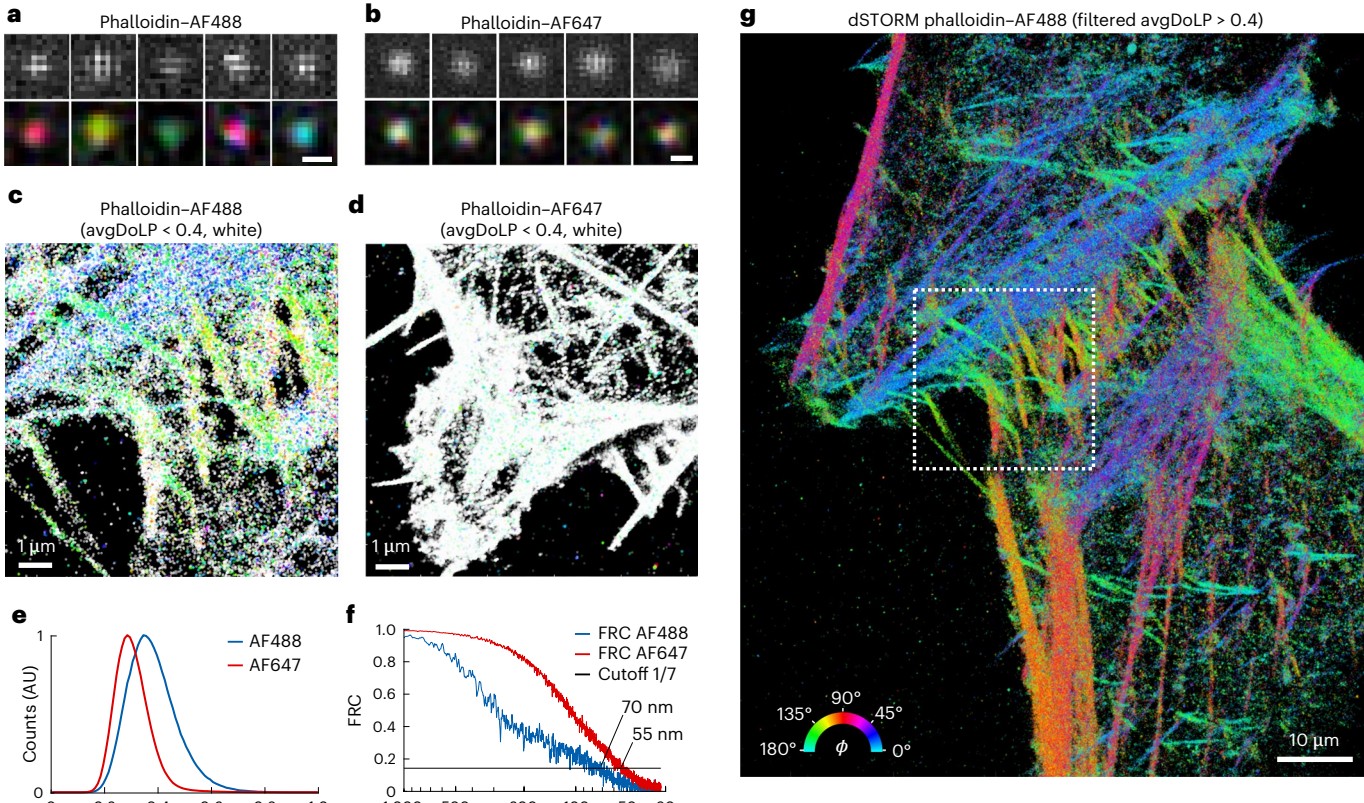

**Fig. 4 | dSTORM imaging of actin in fixed HeLa cells. a**, Representative examples of images of single AF488 molecules from the dSTORM datasets of phalloidin–AF488, unprocessed (top) and as an HSV (hue, AoLP; saturation, DoLP; value, $S_0$) color map (bottom). Scale bar, 300 nm. **b**, The same as in **a**, but for AF647. Scale bar, 300 nm. **c**, POLCAM SMOLM reconstruction of a phalloidin–AF488 dSTORM dataset in the form of a modified $\phi$-color-coded scatterplot. All localizations with an avgDoLP < 0.4 are colored white to indicate that they have high rotational mobility. **d**, The same as **c**, but for phalloidin–AF647. **e**, Comparison of the avgDoLP distribution of single molecules from the phalloidin–AF488 and phalloidin–AF647 datasets. AU, arbitrary units. **f**, FRC curves for the phalloidin–AF488 (70-nm FRC resolution) and phalloidin–AF647 (55-nm FRC resolution) dSTORM datasets. **g**, $\phi$-Color-coded scatterplot of the full dataset from inset **c** (marked by the dotted box), displaying only the points with avgDoLP > 0.4 (that is, localizations with high rotational mobility are not displayed).

acquired using bulk labeling concentrations of different membrane dyes (Di-8-ANEPPS, DiI and Nile red). As expected, the emission dipole moments of Di-8-ANEPPS and Nile red orient perpendicular to the membrane surface, and DiI orients parallel to the membrane surface (Fig. 2e and Supplementary Fig. 33). Nile red was used to collect a point accumulation for imaging in nanoscale topography (PAINT)[4] dataset of a lipid-coated microsphere to generate an experimental accuracy curve for the in-plane angle $\phi$ (Fig. 2f,g).

Experimental orientation estimation and localization precision curves as a function of the number of detected photons were generated using single Alexa Fluor 647 (AF647) dyes immobilized in poly(vinyl alcohol) (PVA). The data points in Fig. 2i are the measured standard deviation on the repeated measurement of the orientation of an AF647 molecule. At 500 detected photons (the default lower threshold on photon number that is used in all datasets), we achieve an experimental in-plane angle precision $\sigma_\phi$ of 7.5° (Fig. 2i), that is, the upper bound on $\sigma_\phi$. The precision converges to a lower bound of 1° at higher photon numbers (Fig. 2i). The experimental localization precision at 500 detected photons is 25–30 nm and converges to 5 nm at high photon numbers (Fig. 2j). The same curves generated using simulations with a realistic noise model largely agree with the experimental data (Supplementary Fig. 35). A complete characterization of the bias and precision on the position, orientation and rotational mobility estimates based on simulations can be found in Supplementary Figs. 36–41. DNA origami with a spacing of 80 nm between binding sites can also be easily resolved using POLCAM (Supplementary Figs. 31 and 32).

## TAB-PAINT imaging of α-synuclein fibrils in vitro

Orientationally resolved imaging of α-synuclein fibrils labeled with the dye Nile red has previously been demonstrated using transient amyloid binding PAINT (TAB-PAINT)[4,65]. Nile red reversibly binds to hydrophobic regions of the fibrils in a defined orientation[12,35]. Therefore, this serves as an excellent test sample for orientation-resolved super-resolution imaging in biologically relevant samples. Fig. 3 shows the POLCAM reconstruction of the α-synuclein fibrils, which were immobilized on a coverglass using poly-L-lysine (PLL) coating and imaged in phosphate-buffered saline (PBS). POLCAM can super-resolve morphologically consistent α-synuclein fibrils with widths of ~50 nm (full-width at half-maximum (FWHM)) over large fields of views of ~50 μm × 50 μm (Fig. 3a,b and Supplementary Fig. 34). When color coding the reconstructions by the in-plane angle $\phi$ estimate, we show that the majority of the Nile red molecules orient parallel to the long axis of the fibril (Fig. 3c–e). The distributions of the measured in-plane angle over short fibril sections have standard deviations around 7° and 9° (Fig. 3f), which approaches the expected precision for this dataset (Fig. 2i; ~5° at 1,000 detected photons). This indicates that the width of this distribution cannot be explained by measurement precision alone and must partly be due to Nile red molecules binding in a range of orientations that are not exactly parallel to the fibril axis.

While a range of avgDoLP and photon values was extracted using POLCAM (Fig. 3g,h), we apply lower-limit filtering thresholds to ensure high-accuracy results: (1) at least 500 detected photons (applied to all single-molecule data presented in this work) and (2) avgDoLP > 0.4

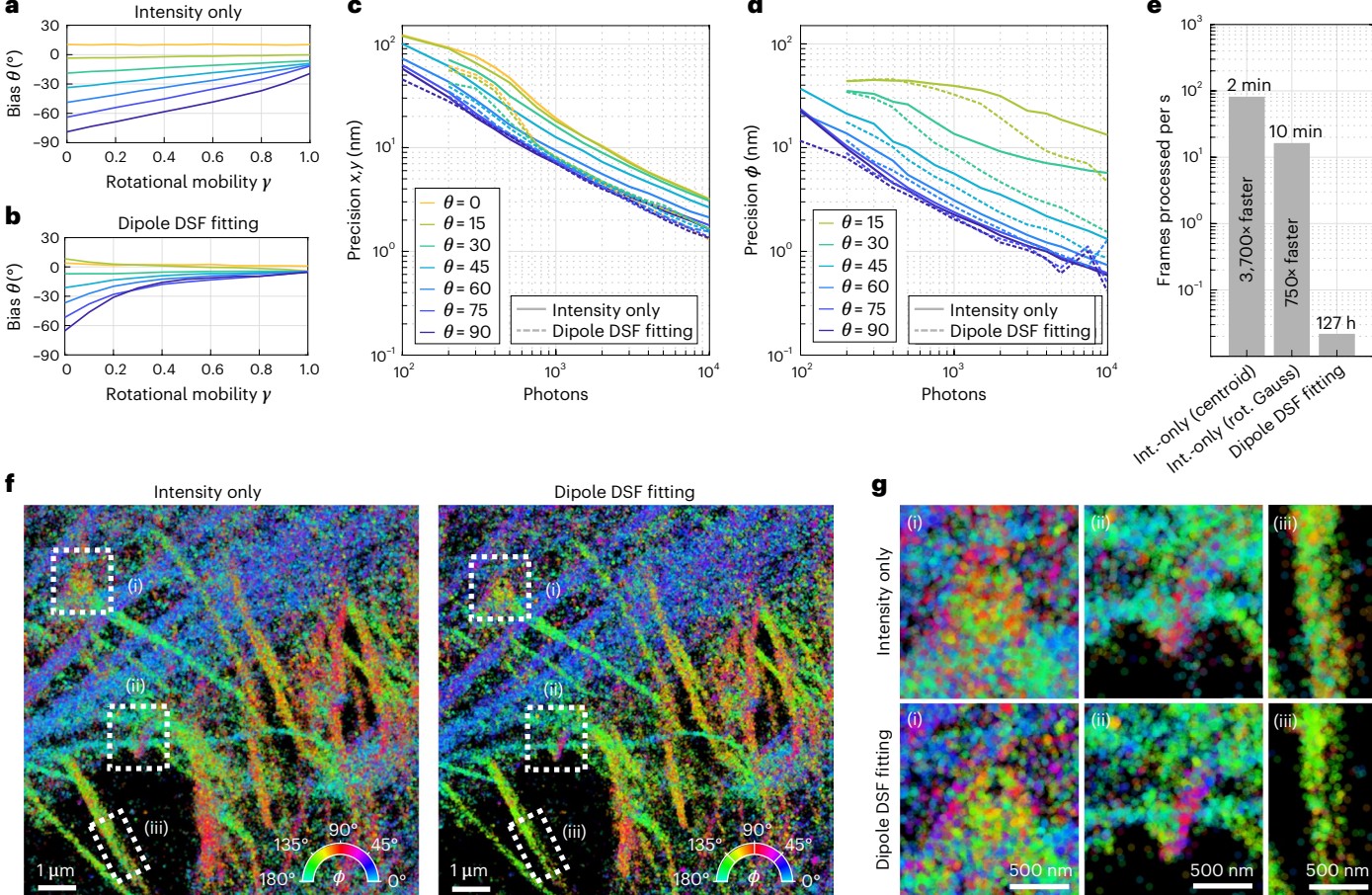

**Fig. 5 | Improving accuracy and precision by considering the DSF shape.**
**a**, The bias on the estimation of the out-of-plane angle $\theta$ as a function of rotational mobility $\gamma$ for the intensity-only algorithm, determined using simulated images of single dipole emitters (using 1,000 photons per emitter and ten background photons per pixel). A rotational mobility parameter $\gamma$ of 0 corresponds to total rotational freedom, and a value of 1 corresponds to perfect rotational immobilization. **b**, The same as **a** but for the DSF-fitting algorithm. **c**, Lateral localization precision as a function of the number of detected photons as determined from simulations (ten background photons per pixel, $\gamma = 1$). Separate curves are shown for molecules at different out-of-plane orientations. **d**, The same as **c** but for the precision of $\phi$ estimation. **e**, The computer processing

time for the datasets from **f** for the intensity (int.)-only algorithm with centroid localization, the intensity-only algorithm with least-squares fitting of a rotated (rot.) asymmetric Gaussian (Gauss) and the DSF-fitting algorithm. Refer to the Methods for computer specifications. **f**, A POLCAM SMOLM reconstruction of a 200 × 200-pixel region from Fig. 4 with 10,000 frames, as analyzed by the intensity-only algorithm (with rotated asymmetric Gaussian fitting) and the DSF-fitting algorithm. Both reconstructions were rendered using approximately the same number of localizations (258,251 localizations for the intensity-only algorithm, 258,172 localizations for the DSF-fitting algorithm). **g**, Insets from the regions in **f** that are marked by dotted boxes. The insets display some structural differences between the reconstructions generated by the two algorithms.

when $\phi$-color-coded data are shown, as localizations with avgDoLP < 0.4 are too rotationally free to estimate a meaningful orientation. An example TAB-PAINT dataset is included in Supplementary Dataset 2.

### dSTORM imaging of actin in fixed HeLa cells
Next, to demonstrate the versatility of POLCAM, we performed SMOLM on eukaryotic cells using direct stochastic optical reconstruction microscopy (dSTORM). Fixed HeLa cells were labeled using phalloidin–Alexa Fluor 488 (AF488) and phalloidin–Alexa Fluor 647 (AF647). In previous literature, this labeling has been shown to result in rotational restriction for AF488 (where the dyes on average align with the axis of actin fibers) and rotational freedom for AF647 (refs. 39,44). Single AF488 and AF647 molecules can be easily detected with POLCAM in the cellular environment, and the difference in rotational mobility between the rotationally constrained AF488 and the randomly oriented AF647 is also directly evident from the images of single molecules (Fig. 4a,b). The difference in rotational mobility between the two labeling approaches is also visible in the resulting super-resolution images (Fig. 4c,d) using a color map in which more orientationally random areas appear white (avgDoLP < 0.4) and the more ordered areas appear colored. Analysis of

the distributions of the avgDoLP revealed an ordered subset of localizations (Fig. 4e) (25% for AF488 and only 5% for AF648). By filtering these localizations using this empirically determined threshold (avgDoLP > 0.4), a refined POLCAM image of actin in cells can be generated that excludes localizations that are too rotationally free to generate an accurate $\phi$ estimate (Fig. 4g). The resolution of the super-resolved images was estimated using Fourier ring correlation (FRC[66]): 70 nm for the AF488 dataset and 55 nm for the AF647 dataset (Fig. 4f).

### Improving accuracy by considering the DSF shape
From equation (2), it is clear that the estimation of the polar angle $\theta$ will become biased in the presence of rotational mobility, as netDoLP will decrease with increasing rotational mobility. Additionally, equation (2) becomes biased in the presence of noise and depends on the refractive index of the sample medium. As a result, unbiased estimation of $\theta$ is more reliably performed by additionally taking the shape of the DSF into account. We adapted the previously published DSF-fitting algorithm RoSE-O[67] for use with a polarization camera. This algorithm fits the shape of the image of a single emitter in all four polarized channels to estimate the orientation and rotational mobility of the emitter. Using

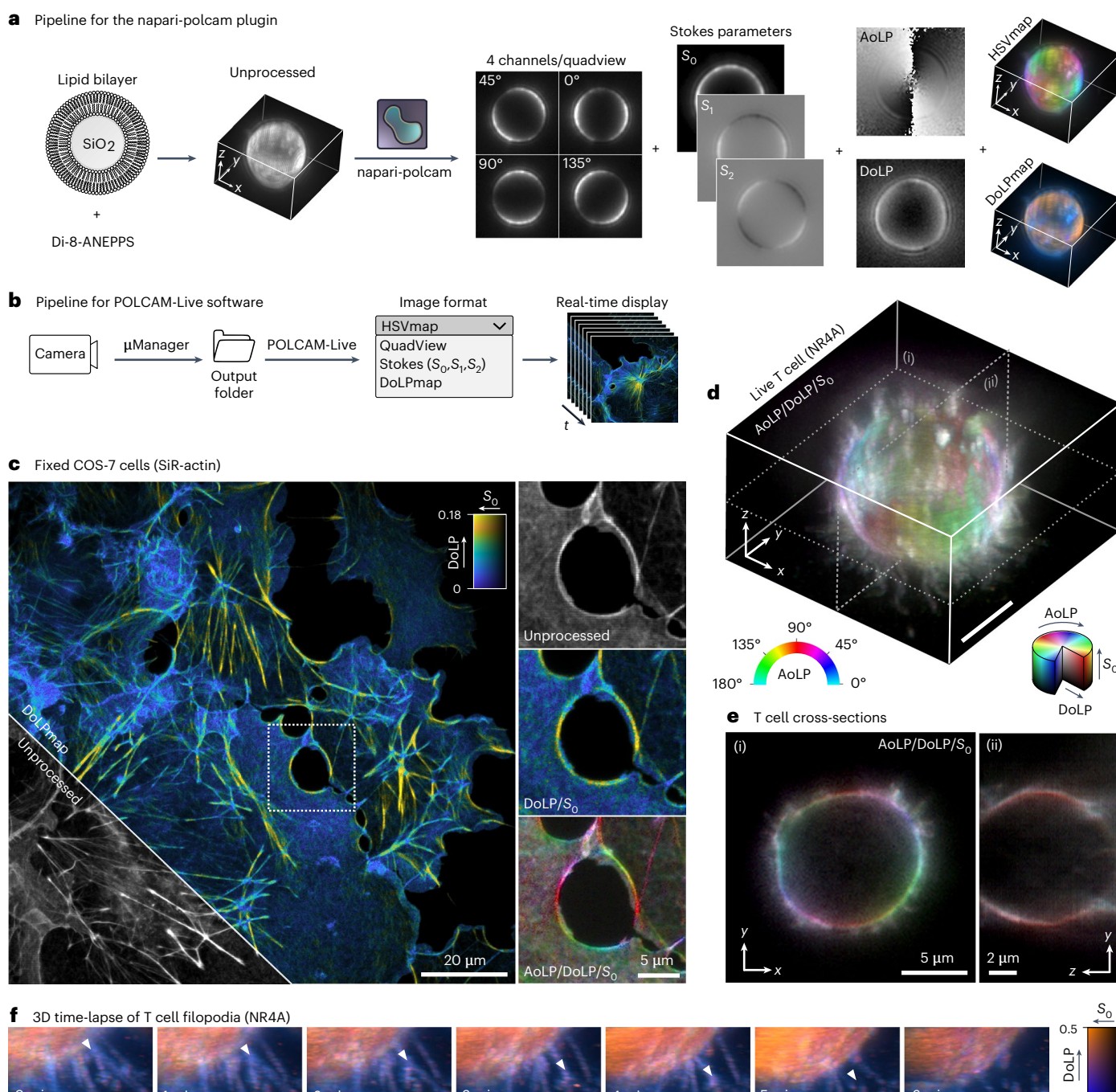

**Fig. 6 | Diffraction-limited polarization microscopy using POLCAM. a**, Image processing workflow of the napari plugin napari-polcam, demonstrated on an example 3D dataset of a lipid bilayer-coated silica microsphere labeled with the membrane dye Di-8-ANEPPS. **b**, Workflow of POLCAM-Live software for real-time processing and rendering of polarization camera images during image acquisition. **c**, Diffraction-limited polarization camera image of fixed COS-7 cells labeled with SiR–actin, rendered using a DoLP color map. An inset marked by a dotted square is shown in an unprocessed, DoLP color map and an HSV polarization color map. **d**, A 3D image of the plasma membrane (using the dye NR4A) of a live T cell. **e**, Two cross-sections of the T cell shown in **d**. **f**, A 3D time-lapse of the movement of the filopodia of a live T cell rendered using a DoLP color map. Randomly polarized regions appear blue (filopodia) and more structured, and therefore polarized areas (larger, more smooth sections of the plasma membrane surface) appear orange. A white triangle tracks the movement of what appears to be a branching point on a filopodium.

simulated images, we compared the performance of the intensity-only algorithm and the DSF-fitting algorithm. As expected, the DSF-fitting algorithm is able to more accurately estimate $\theta$ in the presence of rotational mobility (Fig. 5a,b). For in-plane-oriented emitters, both algorithms achieve the same localization precision across a wide range of signal-to-noise ratios, but the DSF-fitting algorithm outperforms the intensity-only algorithm up to twofold for emitters that are oriented out of plane (Fig. 5c). The precision on the estimation of $\phi$ is similar for in-plane-oriented emitters, but the DSF-fitting algorithm is able to more precisely estimate $\phi$ for more out-of-plane-oriented emitters, especially at high signal-to-noise ratios (Fig. 5d). A complete comparison of the algorithms can be found in Supplementary Figs. 36–41.

Fig. 5f shows super-resolution reconstructions generated by the two algorithms of a subset (200 × 200 pixels and 10,000 frames and

~15 localizations per frame) of the phalloidin–AF488 dSTORM dataset from Fig. 4g. Although both algorithms generate similar-looking reconstructions, zooming in on specific regions (Fig. 5g) seems to confirm that the DSF-fitting algorithm produces a slightly higher-resolution image. A spatial comparison of the $\phi$ estimates shows that the widths of the $\phi$ distributions generated by the DSF-fitting algorithm are slightly more narrow (Supplementary Fig. 41). On a typical workstation (see the Methods for system specifications), the intensity-only algorithm is >750–3,700 times faster (2 minutes for total processing using centroid localization and 10 minutes using least-squares rotated asymmetric Gaussian fitting) than the DSF-fitting algorithm. Therefore, the DSF-fitting algorithm is recommended if the complete 3D orientation ($\phi, \theta$) needs to be estimated. If knowledge of $\phi$ and the rotational mobility proxy avgDoLP is sufficient, the fast intensity-only algorithm is recommended.

### Live diffraction-limited polarization microscopy

The combination of instrumental simplicity and fast computation makes POLCAM compatible with real-time image processing. We developed napari-polcam (a napari plugin for the open-source multidimensional image viewer napari[58] (Fig. 6a)) and a standalone application for on-the-fly processing and rendering called POLCAM-Live (Fig. 6b). Both software take in unprocessed data and convert it into different formats in an easy-to-use interface. The ability to process data live is useful for fast decision making during experiments and alignment (Supplementary Note 8).

We use these software tools to illustrate diffraction-limited polarization imaging, demonstrating 3D time-lapse imaging of the plasma membrane in live human T cells using the new probe NR4A[68]. The 3D images are acquired by axially scanning the objective. The filopodia of the T cells appear more unpolarized (Fig. 6d–f), allowing their simple identification from cell bodies, which is the subject of intense research interest with regard to surface receptor organization[69]. The unpolarized appearance of filopodia is likely due to their small diameter. The 3D motion of the filopodia can be tracked over time (~1 minute per volume, limited by the speed of our z stage). Furthermore, POLCAM can be used in a simplified mode to discriminate ordered versus non-ordered actin structures based on the DoLP. Using a simple DoLP color map, we can distinguish highly ordered regions (yellow) from more disordered regions (blue) in the actin network in COS-7 cells labeled with silicon rhodamine-actin (SiR–actin) (Fig. 6c).

## Discussion

In this work, we present POLCAM, a new method for molecular orientation-resolved fluorescence microscopy that makes use of a polarization camera to dramatically simplify the experimental setup. The method can be used in two modes, SMOLM or diffraction-limited polarization fluorescence microscopy, and can be implemented by simply replacing the conventional detector on a wide-field fluorescence microscope with a polarization camera. The polarization camera used in this work is supported by the popular image acquisition software μManager[57]. Furthermore, we provide software to improve widespread user adaption as well as a comprehensive installation guide and a software user manual.

We present two SMOLM analysis algorithms for POLCAM: (1) a fast algorithm based on Stokes parameter estimation and simple intensity measurements for robust in-plane angle estimation compatible with high-throughput data collection and (2) an unbiased 3D orientation estimation algorithm that fits a DSF model. Due to the computational cost of the DSF-fitting algorithm, it is more suited to the analysis of small datasets.

In conventional four-channel polarized detection, where beam splitters and polarization optics are used to separate the fluorescence into four channels, it is more likely that channel-dependent aberrations will occur. Due to the simplicity of the experimental setup of POLCAM, this is avoided, simplifying the use of a DSF-fitting algorithm. Moreover, there is no need for channel registration and localization grouping because single-molecule localization is performed on the estimated incident intensity ($S_0$) on the micropolarizer array, meaning that all in-plane orientations are equally detectable and no localizations are therefore missed. A diffraction-limited polarization color map (HSVmap) can be generated and displayed in real-time during experiments, for which we provide the standalone software POLCAM-Live. In theory, the intensity-only SMOLM algorithm is also fast enough to be compatible with real-time processing, as a 200 × 200-pixel image (10 × 10 μm) with ~15 localizations takes about 10 ms to process, which is less than typical camera exposure times. This could enable molecular orientation event-triggered microscopy[70,71], something that would be extremely challenging if DSF fitting is required.

Because a polarization camera uses polarizers to achieve polarized detection, on average, the system is 50% efficient, as half of the photons that are captured by the objective will be absorbed by polarizers. Nevertheless, we demonstrate in this work that this approach is compatible with a wide variety of fluorophores (for example, AF488, Nile red, SiR, Cy5, SYTOX Orange). The current generation of polarization cameras lacks the sophistication of modern scientific complementary metal–oxide–semiconductor (sCMOS) cameras in areas such as the quantum efficiency of the detector (70% versus 95%) and the presence of onboard denoising algorithms. We expect this to improve over time. Further improvements to the analysis approach can be made by performing a pixel-dependent characterization of any defects of the micropolarizer array or deviation from an ideal micropolarizer array[72].

The success of a molecular orientation-resolved experiment stands and falls with the labeling approach. It would thus be meaningless to attempt to measure molecular orientation if the relative orientation of a target and the probe is random. Therefore, the use of antibodies with multiple dyes conjugated at random orientations (for example, random lysine labeling) or staining using secondary antibodies is likely not suitable. Currently, the number of labeling protocols that restrict or control the orientation and rotational mobility of a fluorophore with respect to their target is still limited, and there is a strong need for the development of more labeling approaches and the discovery of suitable probes. Examples are the use of bifunctional rhodamines[16] and the genetically engineered rigid protein linker POLArIS[73]. It is likely that many common fluorescent probes are suitable for molecular orientation-resolved microscopy, but they have simply never been evaluated for this purpose. We anticipate that, because of the accessibility and compatibility with high-throughput data acquisition, POLCAM will accelerate this much-needed development and discovery of new probes and expand the current toolkit to cover more biological systems.

We note that polarization cameras can also be used for label-free microscopy[74–77], leaving the possibility for multiplexing of fluorescence and label-free techniques. Overall, we envisage that the combination of POLCAM's simple implementation and ease of use, computational speed and open-source software will lead to new biological insight across diverse systems.

## Online content

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

## Methods

### Optical setups

Experiments were performed on three very similar wide-field fluorescence microscopes. SYTOX Orange, AF647 immobilized in polymer, TAB-PAINT of α-synuclein fibrils and diffraction-limited polarization imaging of the actin network in COS-7 cells were performed on 'microscope 1'. The dSTORM experiments were performed on 'microscope 2'. The lipid bilayer-coated silica microsphere and live T cell experiments were performed on 'microscope 3'.

'Microscope 1' is a wide-field fluorescence microscope (Eclipse Ti-U, Nikon), with illumination entering the microscope body through the back illumination port. The beams from three free-space lasers (515 nm, 150 mW, Spectra-Physics; 532 nm, 120 mW, Odic Force Lasers; 638 nm, 350 mW, Odic Force Lasers) were expanded, spectrally and spatially filtered, combined with dichroics and focused to a spot in the back focal plane of an oil-immersion objective (Plan Apo, ×60 NA-1.40 oil, DIC H, ∞/0.17, WD 0.21, Nikon) using an achromatic doublet lens (AC254-300-A, Thorlabs). This lens and a periscope were mounted on a linear translation stage to allow manual adjustment of the beam emerging from the objective and switching between EPI, HILO and TIRF illumination. Approximately circular polarization at the sample plane was achieved using quarter waveplates (WPQ10M-514, WPQ10M-532 and WPQ10M-633, Thorlabs). For diffraction-limited imaging of the actin network in COS-7 cells, a different laser source was used with a multimode fiber (LDI-7 Laser Diode Illuminator, 89 North). Fluorescence was filtered by a dichroic beam splitter (Di03-R405/488/532/635-t1 for AF647 and Di03-R532-t1 for SYTOX Orange and Nile red; Semrock) and emission filters (BLP01-532R for SYTOX Orange, BLP01-635R for AF647, BLP01-532R and FF01-650/200 for Nile red; Semrock). The fluorescence was focused on a polarization camera (CS505MUP, Thorlabs) that was placed directly at the microscope body camera port. The pixel size of the camera is 3.45 μm × 3.45 μm, resulting in a virtual pixel size of 57.5 × 57.5 nm. The microscope PC was a Dell OptiPlex 7070 Mini Tower running on Windows 10 (64 bit) with an Intel Core i9-9900 processor and 32 GB of RAM.

'Microscope 2' is functionally similar to 'microscope 1', with a different microscope body (Eclipse Ti-E, Nikon) and laser source (Omicron LightHUB with 405-, 488-, 561- and 638-nm lasers, single-mode fiber, collimator RC08APC-P01, Thorlabs) and a ×60 1.42-NA oil-immersion objective (Olympus, PlanApo N). A quadband imaging dichroic (Di03-R405/488/532/635-t1, Semrock) and emission filters were used (BLP01-488R and FF01-582/64 for AF488, BLP01-635R for AF647; Semrock). The fluorescence was focused on a polarization camera (CS505MUP, Thorlabs) that was placed directly at the microscope body camera port, resulting in a virtual pixel size of 51.7 × 51.7 nm. Note that an Olympus objective (assuming use of an Olympus body with a 180.0-mm focal length tube lens) was used with a Nikon body (200.0-mm focal length tube lens).

'Microscope 3' is also functionally similar to 'microscope 1', other than the microscope body (Eclipse Ti-E, Nikon), lasers (Cobalt C-FLEX combiner with 405-, 488-, 515-, 561- and two 638-nm lasers, free space) coupled into a square-core multimode fiber (M97L02, Thorlabs) with a custom vibration motor-based mode scrambler and a ×100 1.49-NA oil-immersion objective. A 4f system consisting of two achromatic lenses (AC254-050-A-ML and AC254-100-A-ML, Thorlabs) was included in the emission path to demagnify the image 2×, resulting in a total system magnification of ×50 and thus a virtual pixel size of 69 × 69 nm. Imaging dichroics (Di03-R515-t1 for NR4A and Nile red and Di03-R405/488/532/635-t1 for SiR–actin; Semrock) and emission filters (BLP01-532R and FF01-650/200 for NR4A, BLP01-635R for SiR–actin; Semrock) were used. A multimode fiber was used for all diffraction-limited imaging experiments, as this allowed us to achieve highly randomized polarization at the sample plane (compared to using a quarter-wave plate), resulting in negligible photoselection.

### Simulations

The emission of a fluorescent molecule was modeled as the far field of an oscillating electric dipole as previously described in ref. 20. Transmission of the emission through the micropolarizer array on the camera sensor was modeled for each camera pixel using a Jones matrix $J_{LP}$ for a linear polarizer with an axis of transmission at an angle $\eta$ from the $x$ axis[78]:

$$\mathbf{E}' = J_{LP}\mathbf{E} \tag{6}$$

$$J_{LP} = \begin{pmatrix} \cos^2\eta & \cos\eta\sin\eta \\ \cos\eta\sin\eta & \sin^2\eta \end{pmatrix}. \tag{7}$$

Unless otherwise specified, the following system parameters were used: emission wavelength of 630 nm, ×60 oil-immersion objective with an NA of 1.4, a tube lens with a focal length of 200.0 mm, physical camera pixel size of 3.45 μm. The molecules are placed on a glass–water refractive index interface ($n_{glass} = n_{oil} = 1.518$, $n_{water} = 1.33$) and in focus. A CMOS camera noise model was used (Supplementary Note 1.7).

### Image acquisition

On all instruments, image acquisition was performed using μManager[57] (μManager version 2.0.0, http://micro-manager.org, RRID: SCR_000415) or ThorCam (ThorCam version 3.6.0, https://www.thorlabs.com/software_pages/ViewSoftwarePage.cfm?Code=ThorCam). Data were always recorded in unprocessed format. All experiments on all three microscopes were performed using epi-illumination (with the exception of the dSTORM experiment where a steep HILO angle was used) to easily control the polarization state at the sample plane. Perfectly random or circular polarization at the sample plane is very challenging to achieve experimentally; therefore, to some degree, there will always be a dominant axis of polarization. Transitioning from EPI to HILO and TIRF will change the amount of photoselection at the sample plane, as the dominant axis and degree of randomness or ellipticity of the polarization will change in a way that is challenging to quantify and reproduce. Before each experiment and/or change of inclination of the excitation beam, the polarization of the excitation beam at the sample is tuned by rotating a quarter-wave plate. Alignment is deemed optimal when the DoLP of the background is minimized (or the appearance of gridding distinctive of polarized background disappears). This step is not necessary if lasers are coupled into a multimode fiber. The imaging parameters for the data shown in all main figures and supplementary figures are summarized in Supplementary Table 3.

### Image analysis

Single-molecule data analysis was performed using the MATLAB (MATLAB R2022a, MathWorks, http://www.mathworks.com/products/matlab/, RRID:SCR_001622) application POLCAM-SR, which includes tools for localization, filtering, drift correction and data visualization. The source code and installer are available on GitHub at https://github.com/ezrabru/POLCAM-SR. Diffraction-limited data analysis was performed using POLCAM-SR and a napari[58] plugin called napari-polcam for processing and visualization of multidimensional polarization camera datasets. The source code and installation instructions for napari-polcam are available on GitHub at https://github.com/ezrabru/napari-polcam. Refer to Supplementary Note 7 for a detailed description of the image analysis pipeline.

### SYTOX Orange on coverglass

Glass coverslips (VWR Collection, 631-0124) were cleaned with argon plasma for 30 min (Expanded Plasma Cleaner, PDC-002, Harrick Plasma). An imaging chamber was created on the coverslips using Frame-Seal slide chambers (9 × 9 mm, SLF0201, Bio-Rad). The glass in the chamber was coated with 70 μl PLL (0.01% (wt/vol), P4707,

Sigma-Aldrich) for 15 min. After removing excess PLL and washing three times with filtered PBS (0.02-µm syringe filter, Whatman, 6809-1102), 50 µl of 1 nM SYTOX Orange (S11368, Invitrogen) was added gently. The sample was imaged immediately.

This protocol is available on https://www.protocols.io as 'Imaging single SYTOX Orange molecules on a PLL-coated cover glass'[79].

## PAINT imaging of lipid bilayer-coated silica microspheres

To prepare lipid bilayer-coated silica microspheres, a slightly modified version of the protocol in ref. 33 was used. First, lipid vesicles of a certain composition were prepared as follows: DPPC (850355C, Avanti Polar Lipids) and cholesterol (C8667-5G, Sigma-Aldrich) were dissolved in chloroform (366927, Sigma-Aldrich) to (respectively) 25 mg ml$^{-1}$ and 10 mg ml$^{-1}$. A DPPC–40% cholesterol mixture was prepared by combining 23 µl DPPC and 20 µl cholesterol. The solvent was evaporated overnight under vacuum. The lipid–cholesterol mixture was rehydrated using 1 ml Tris-Ca$^{2+}$ buffer (100 mM NaCl, 3 mM CaCl$_2$, 10 mM Tris base, pH 7.4) and vortexed for 30 s. The solution was sonicated using a tip sonicator (cycles of 45 s on, 15 s off, 60% amplitude) for 40 min until the solution ran clear. The sonicated solution was centrifuged for 90 s at 14,000 rcf to remove titanium residue from the sonicator probe.

Next, silica microspheres with a diameter of 5 µm (44054-5ML-F, Sigma-Aldrich) were diluted to approximately 2.8 mg ml$^{-1}$ and cleaned by centrifuging and replacing the stock buffer with Tris-Ca$^{2+}$. The microspheres and lipid vesicle solutions were heated to 65 °C using a heated water bath and mixed together in a 1:1 ratio. After 30 min at 65 °C, the mixtures were slowly cooled down to room temperature (by turning the heating bath off). The buffer was gradually replaced with Tris (100 mM NaCl, 10 mM Tris base, pH 7.4) by centrifugation (5 min at 0.3 rcf) and replacement of two-thirds of the supernatant with Tris, repeated six times. The lipid-coated microspheres were stored at 4 °C and used within less than 2 weeks of preparation.

For imaging, the lipid-coated microspheres were added to an argon plasma-cleaned, PLL-coated coverglass (VWR Collection, 631-0124), and 1 nM NR4A or Nile red was added for PAINT imaging of the lipid bilayer. The Nile red derivative NR4A was provided by A.S. Klymchenko at the Université de Strasbourg. For diffraction-limited imaging of lipid-coated microspheres, 100 nM dye in PBS was used (for Nile red, NR4A and Di-8-ANEPPS). All buffers (PBS, Tris, Tris-Ca$^{2+}$) were filtered before use (0.02-µm syringe filter, Whatman, 6809-1102).

This protocol is available on https://www.protocols.io as 'Preparation and imaging of lipid bilayer-coated silica microspheres'[80].

## AF647 immobilized in PVA

A PVA solution (1%, 1 g in 100 ml) was prepared by slow addition of solid PVA into filtered (0.02-µm syringe filter, Whatman, 6809-1102) Milli-Q water with stirring. The solution was then heated to 90 °C, stirred for 30 min and then removed from heat with continued stirring for 12 h. The PVA solution was then filtered (0.02-µm syringe filter, Whatman, 6809-1102) and stored at 4 °C. AF647 (500 pM) was diluted in the 1% PVA solution, and 10 µl was then spin-cast (3,000 rpm, 45 s) onto a glass coverslip cleaned with Ar plasma (ODC-002, Harrick Plasma) and sealed before imaging.

This protocol is available on https://www.protocols.io as 'Imaging single AF647 molecules immobilised in PVA on a cover glass'[81].

## TAB-PAINT of α-synuclein fibrils

To prepare α-synuclein fibrils, α-synuclein monomer was diluted to a concentration of 70 µM in PBS (with 0.01% NaN$_3$) and incubated at 37 °C in a shaker (200 rpm) to aggregate for >24 h. To prepare fibrils for imaging, glass coverslips (VWR Collection, 631-0124) were plasma cleaned for 1 h (argon plasma cleaner, PDC-002, Harrick Plasma). An imaging chamber was created on the coverslips using Frame-Seal slide chambers (9 × 9 mm, SLF0201, Bio-Rad). The glass in the chamber was coated with 70 µl PLL (0.01% (wt/vol), P4707, Sigma-Aldrich) for 30 min. After removing excess

PLL and washing three times with filtered PBS (20-nm pore filters), 50 µl TetraSpeck beads (0.1 µM stock, 10× diluted) were added for lateral drift correction. Samples were washed again 3× with filtered PBS and 50 µl α-synuclein fibrils (diluted to 35 mM monomer concentration from a 70 mM monomer concentration stock that was stored at 4 °C). Fibrils were stuck to PLL by pipetting up and down a couple of times in the four corners of the chamber. Before imaging, excess solution was removed, followed by a gentle wash with filtered PBS. Imaging buffer (50 µl, 1 nM Nile red in PBS, diluted from a 1 mM aliquot in DMSO, stored at −20 °C) was added, and the sample was imaged immediately.

This protocol is available on https://www.protocols.io as 'TAB-PAINT imaging of alpha-synuclein fibrils using Nile Red'[82].

## dSTORM of actin in fixed HeLa cells

HeLa TDS cells (RRID:CVCL_0030) were cultured in DMEM medium (Gibco, Invitrogen) supplemented with 10% FBS (Life Technologies), 1% penicillin–streptomycin (Life Technologies) and 1% glutamine (Life Technologies) at 37 °C with 5% CO$_2$. Cells were periodically tested for mycoplasma contamination and passaged three times per week. Cells were plated at low density on high-precision glass coverslips (MatTek, P35G-0.170-14-C) 1 d before fixation for dSTORM experiments.

Cells were simultaneously fixed and permeabilized in cytoskeleton buffer (10 mM MES, 138 mM KCl, 3 mM MgCl$_2$, 2 mM EGTA, 4.5% sucrose (wt/vol), pH 7.4) with 4% paraformaldehyde and 0.2% Triton for 6 min at 37 °C and further fixed in cytoskeleton buffer with 4% paraformaldehyde for 14 min at 37 °C. After fixation, cells were washed three times with PBST (PBS supplemented with 0.1% Tween) and permeabilized a second time in PBS with 0.5% Triton for 5 min at room temperature. The samples were then washed three times with PBST and blocked for 30 min with 5% BSA. Cells were washed three times with PBST and then incubated with AF488–phalloidin (A12379, Invitrogen, 1:50 in PBS) for 1 h in the dark, followed by two washes with PBS. Before dSTORM imaging, PBS was replaced with dSTORM imaging buffer (base buffer consisting of 0.56 M glucose, 50 mM Tris (pH 8.5) and 10 mM NaCl supplemented with 5 U ml$^{-1}$ pyranose oxidase (Sigma, P4234), 10 mM cysteamine (Sigma, 30070), 40 µg ml$^{-1}$ catalase (Sigma, C100) and 2 mM cyclooctatetraene (Sigma, 138924).

This protocol is available on https://www.protocols.io as 'dSTORM of actin in fixed HeLa cells'[83].

## Fixed COS-7 cells labeled with SiR–actin

A commercial slide (GATTA-Cells 4C, GATTAquant) was used, containing fixed COS-7 cells (RRID:CVCL_0224) labeled with SiR–actin (CY-SC001, Cytoskeleton).

## Live cell imaging of the plasma membrane of Jurkat T cells

J8 LFA-1 cells were incubated overnight (~18 h) in complete RPMI (StableCell RPMI-1640 medium, Sigma) supplemented with 10% (vol/vol) fetal calf serum, 1% (vol/vol) HEPES buffer and 1% (vol/vol) penicillin–streptomycin antibiotics. Cells (1 ml) were collected by centrifugation and resuspended in phenol red-free RPMI supplemented with 1% HEPES.

Round coverslips were rinsed with IPA and Milli-Q, dried and cleaned with Ar plasma for 20 min. Grace Bio-Labs CultureWells were attached, and the slide was incubated with OKT3 antibody (provided by the Human Immunology Unit, WIMM) for 30 min. The slide was washed five times with phenol red-free RPMI supplemented with 1% HEPES, and a final wash was performed with phenol red-free RPMI supplemented with 1% HEPES and 200 nM NR4A (MemGlow NR4A Membrane Polarity Probe, MG06, Cytoskeleton) before imaging.

This protocol is available on https://www.protocols.io as 'Live-cell imaging of the plasma membrane of Jurkat T cells'[84].

## Reporting summary

Further information on research design is available in the Nature Portfolio Reporting Summary linked to this article.

## Data availability

The datasets generated as part of this study were uploaded to Zenodo: pixel-dependent camera calibration results (https://doi.org/10.5281/zenodo.10578307)[85], single SYTOX Orange on a coverglass (https://doi.org/10.5281/zenodo.10469322)[86], PAINT data of single Nile red dye molecules binding to lipid bilayer-coated silica microspheres (https://doi.org/10.5281/zenodo.10469444)[87], TAB-PAINT data (https://doi.org/10.5281/zenodo.10470795)[88], dSTORM phalloidin–AF488 (https://doi.org/10.5281/zenodo.10470982)[89], dSTORM phalloidin–AF647 (https://doi.org/10.5281/zenodo.10732697)[90] and T cells (https://doi.org/10.5281/zenodo.10471496)[91].

## Code availability

The installers and source code of all custom code and software used in this work are available on GitHub and Zenodo: POLCAM-SR (https://github.com/ezrabru/POLCAM-SR, https://doi.org/10.5281/zenodo.10732422 (ref. 92), RRID:SCR_025343), POLCAM-Live (https://github.com/ezrabru/POLCAM-Live, https://doi.org/10.5281/zenodo.10732437 (ref. 93), RRID:SCR_025342), napari-polcam (https://github.com/ezrabru/napari-polcam, https://doi.org/10.5281/zenodo.10732441 (ref. 94), RRID:SCR_025341), RoSE-O_POLCAM (https://github.com/Lew-Lab/RoSE-O_polCam, RRID:SCR_025340) and the camera calibration software (https://github.com/TheLeeLab/cameraCalibrationCMOS, https://doi.org/10.5281/zenodo.10732469 (ref. 95), RRID:SCR_025339).

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

## Acknowledgements

This research was funded in part by Aligning Science Across Parkinson's ASAP-000509 through the Michael J. Fox Foundation for Parkinson's Research. For the purpose of open access, we have applied a CC BY public copyright license to all author accepted manuscripts arising from this submission. R.P. was funded by the European Research Council (ERC Consolidator Grant 820188-NanoMechShape to E.K.P.). We thank the Cambridge Advanced Imaging Centre (University of Cambridge) for the use of a microscope for the dSTORM experiments and the mechanical workshop (Y. Hamied, Department of Chemistry, University of Cambridge) for building custom parts of optical setups used in this work. We thank A. Ponjavic and J. Beckwith for critical reading of the manuscript. We thank L. Elfeky for testing the POLCAM-SR software.

## Author contributions

S.F.L. conceived the project. S.F.L., K.O.'H. and M.D.L. supervised the research. E.B. built the optical setups. E.B. acquired data. E.B. programmed and performed simulations. E.B. and O.Z. analyzed data. E.B. programmed the POLCAM-SR, POLCAM-Live and napari-polcam software. O.Z. adapted the RoSE-O algorithm for polarization cameras. L.-M.N. prepared the α-synuclein fibrils and dye in polymer samples. R.P. prepared the dSTORM actin samples. M.K. prepared the live T cell samples. E.B., M.C. and S.D. prepared the lipid bilayer-coated microsphere samples. T.W. provided the protocol for preparing lipid bilayer-coated microspheres. A.S.K. provided the dye NR4A. S.J.D. provided the J8 LFA-1 cell line and OKT3 antibodies. E.B. and S.F.L. wrote the manuscript with input from all authors.

## Competing interests

The authors declare no competing interests.

## Additional information

**Correspondence and requests for materials** should be addressed to Steven F. Lee.

# Reporting Summary

## Statistics

For all statistical analyses, confirm that the following items are present in the figure legend, table legend, main text, or Methods section.

| n/a | Confirmed | |
|---|---|---|
| ☐ | ☒ | The exact sample size (*n*) for each experimental group/condition, given as a discrete number and unit of measurement |
| ☐ | ☒ | A statement on whether measurements were taken from distinct samples or whether the same sample was measured repeatedly |
| ☒ | ☐ | The statistical test(s) used AND whether they are one- or two-sided<br>*Only common tests should be described solely by name; describe more complex techniques in the Methods section.* |
| ☒ | ☐ | A description of all covariates tested |
| ☒ | ☐ | A description of any assumptions or corrections, such as tests of normality and adjustment for multiple comparisons |
| ☐ | ☒ | A full description of the statistical parameters including central tendency (e.g. means) or other basic estimates (e.g. regression coefficient) AND variation (e.g. standard deviation) or associated estimates of uncertainty (e.g. confidence intervals) |
| ☒ | ☐ | For null hypothesis testing, the test statistic (e.g. *F*, *t*, *r*) with confidence intervals, effect sizes, degrees of freedom and *P* value noted<br>*Give P values as exact values whenever suitable.* |
| ☒ | ☐ | For Bayesian analysis, information on the choice of priors and Markov chain Monte Carlo settings |
| ☒ | ☐ | For hierarchical and complex designs, identification of the appropriate level for tests and full reporting of outcomes |
| ☒ | ☐ | Estimates of effect sizes (e.g. Cohen's *d*, Pearson's *r*), indicating how they were calculated |

*Our web collection on statistics for biologists contains articles on many of the points above.*

## Software and code

Policy information about availability of computer code

| | |
|---|---|
| Data collection | The open-source software Micro-Manager (version 2.0.0) was used for microscope control and data acquisition, and downloaded from https://micro-manager.org/Download_Micro-Manager_Latest_Release. A Micro-Manager device adapter for the polarization camera used in this work was included as part of the free image acquisition software ThorCam (v3.6.0, Thorlabs), which was downloaded from https://www.thorlabs.com/software_pages/ViewSoftwarePage.cfm?Code=ThorCam. |
| Data analysis | Basic image inspection and image cropping was performed in ImageJ (v1.53t). Single-molecule data was analyzed using custom MATLAB (version R2022a, Mathworks) application POLCAM-SR for which the source code and installer are available on GitHub at https://github.com/ezrabru/POLCAM-SR. Diffraction-limited, high-dimensional polarisation camera image processing and visualization was performed using a custom plugin napari-polcam for the open source software napari (v0.4.17), installed using the instructions at https://napari.org/stable/. The source code for the plugin is available at https://github.com/ezrabru/napari-polcam. |

For manuscripts utilizing custom algorithms or software that are central to the research but not yet described in published literature, software must be made available to editors and reviewers. We strongly encourage code deposition in a community repository (e.g. GitHub). See the Nature Portfolio guidelines for submitting code & software for further information.

March 2021

## Data

Policy information about availability of data

All manuscripts must include a data availability statement. This statement should provide the following information, where applicable:

- Accession codes, unique identifiers, or web links for publicly available datasets
- A description of any restrictions on data availability
- For clinical datasets or third party data, please ensure that the statement adheres to our policy

The datasets generated as part of this study were uploaded to Zenodo: pixel-dependent camera calibration results (\url{https://doi.org/10.5281/zenodo.10578307}), single SYTOX Orange on a cover glass (\url{https://doi.org/10.5281/zenodo.10469322}), PAINT data of single Nile red dyes binding to lipid bilayer-coated silica microspheres (\url{https://doi.org/10.5281/zenodo.10469444}), TAB-PAINT data (\url{https://doi.org/10.5281/zenodo.10470795}), dSTORM phalloidin-AF488 (\url{https://doi.org/10.5281/zenodo.10470982}), dSTORM phalloidin-AF647 (\url{https://doi.org/10.5281/zenodo.10732697}) and T cells (\url{https://doi.org/10.5281/zenodo.10471496}).

## Human research participants

Policy information about studies involving human research participants and Sex and Gender in Research.

| | |
|---|---|
| Reporting on sex and gender | n.a. |
| Population characteristics | n.a. |
| Recruitment | n.a. |
| Ethics oversight | n.a. |

Note that full information on the approval of the study protocol must also be provided in the manuscript.

# Field-specific reporting

Please select the one below that is the best fit for your research. If you are not sure, read the appropriate sections before making your selection.

☒ Life sciences ☐ Behavioural & social sciences ☐ Ecological, evolutionary & environmental sciences

For a reference copy of the document with all sections, see nature.com/documents/nr-reporting-summary-flat.pdf

# Life sciences study design

All studies must disclose on these points even when the disclosure is negative.

| | |
|---|---|
| Sample size | All experiments used in this study have been performed at least 3 times, to assure results generated by the presented method are repeatable. |
| Data exclusions | Localisations with less than 500 detected photons were excluded from all single-molecule datasets in this study, to avoid biased orientation estimates (as explained in the manuscript). |
| Replication | Each experiment presented in this work was replicated successfully at least 3 times to assure repeatability of the results. Single dye on glass or in polymer was replicated successfully >10 times on different optical setups, days, sample regions and different dyes. PAINT imaging of lipid-coated glass microspheres was successfully repeated >3 times on separate days and sample regions. TAB-PAINT imaging of alpha-synuclein fibrils was successfully repeated >3 times on different optical setups, days and sample regions. dSTORM imaging of the actin network of HeLa cells was successfully repeated 3 times (3 cells per labelling method). Live T cell imaging was successfully repeated 3 times on separate days and different sample regions. |
| Randomization | Randomization is not relevant to this study. |
| Blinding | Blinding is not relevant to this study. |

# Reporting for specific materials, systems and methods

We require information from authors about some types of materials, experimental systems and methods used in many studies. Here, indicate whether each material, system or method listed is relevant to your study. If you are not sure if a list item applies to your research, read the appropriate section before selecting a response.

## Materials & experimental systems

| n/a | Involved in the study |
|---|---|
| ☐ | ☒ Antibodies |
| ☐ | ☒ Eukaryotic cell lines |
| ☒ | ☐ Palaeontology and archaeology |
| ☒ | ☐ Animals and other organisms |
| ☒ | ☐ Clinical data |
| ☒ | ☐ Dual use research of concern |

## Methods

| n/a | Involved in the study |
|---|---|
| ☒ | ☐ ChIP-seq |
| ☒ | ☐ Flow cytometry |
| ☒ | ☐ MRI-based neuroimaging |

## Antibodies

| | |
|---|---|
| Antibodies used | OKT3 antibody (provided by the Human Immunology Unit, WIMM, Oxford, UK) |
| Validation | Unknown |

## Eukaryotic cell lines

Policy information about cell lines and Sex and Gender in Research

| | |
|---|---|
| Cell line source(s) | HeLa cells: derived from cells isolated from the cervix of a 31-year-old female with adenocarcinoma. COS7 cells: derived from the CV-1 cell line (ATCC CCL-70) by transformation with an origin defective mutant of SV40 which codes for wild type T antigen. This is an African green monkey kidney fibroblast-like cell line. Jurkat T cells: clone of the Jurkat-FHCRC cell line (ATCC TIB-152), established from the peripheral blood of a 14-year-old, male, acute T-cell leukemia patient. |
| Authentication | The cell lines used were not authenticated. |
| Mycoplasma contamination | HeLa cells and Jurkat T cells are periodically tested for mycoplasma contamination and tested negative. It is unknown whether COS7 cells were tested for mycoplasma contamination. |
| Commonly misidentified lines (See ICLAC register) | n.a. |

