## [Peer Review File · Nature Methods]

Peer Review Information

Manuscript Title: POLCAM: Instant molecular orientation microscopy for the life sciences

Corresponding author name(s): Steven Lee

Editorial Notes:

Reviewer Comments & Decisions:

Decision Letter, initial version:

Date: 15th May 23 16:48:41

Last Sent: 15th May 23 16:48:41

Triggered By: Rita Strack

From: rita.strack@us.nature.com

To: sl591@cam.ac.uk

CC: methods@us.nature.com

Subject: Decision on Nature Methods submission NMETH-A51455

Message: 15th May 2023

Dear Steven,

Please let me begin by apologizing for the duration of this review. We were waiting on the third referee.

Your Article, "POLCAM: Instant molecular orientation microscopy for the life sciences", has now been seen by three reviewers. As you will see from their comments below, although the reviewers find your work of considerable potential interest, they have raised a number of concerns. We are interested in the possibility of publishing your paper in Nature Methods, but would like to consider your response to these concerns before we reach a final decision on publication.

We therefore invite you to revise your manuscript to address these concerns, emphasizing the technical correctness of your approach and carefully assessing achieved/achievable image quality.

Please note, we do not require a Napari plug-in to be developed. Though you may want to mention this as a possible future direction.

[REDACTED]

We hope to receive your revised paper within three months. If you cannot send it within this time, please let us know. In this event, we will still be happy to reconsider your paper at a later date so long as nothing similar has been accepted for publication at Nature Methods or published elsewhere.

OPEN SCIENCE REQUIREMENTS

REPORTING SUMMARY AND EDITORIAL POLICY CHECKLISTS

Reporting summary: <https://www.nature.com/documents/nr-reporting-summary.zip>
Editorial policy checklist: <https://www.nature.com/documents/nr-editorial-policy-checklist.zip>

If your paper includes custom software, we also ask you to complete a supplemental

reporting summary.

DATA AVAILABILITY

All novel DNA and RNA sequencing data, protein sequences, genetic polymorphisms, linked genotype and phenotype data, gene expression data, macromolecular structures, and proteomics data must be deposited in a publicly accessible database, and accession codes and associated hyperlinks must be provided in the "Data Availability" section.

Please include a "Data availability" subsection in the Online Methods. This section should inform readers about the availability of the data used to support the conclusions of your study, including accession codes to public repositories, references to source data that may be published alongside the paper, unique identifiers such as URLs to data repository entries, or data set DOIs, and any other statement about data availability. At a minimum, you should include the following statement: "The data that support the findings of this study are available from the corresponding

author upon request”, describing which data is available upon request and mentioning any restrictions on availability. If DOIs are provided, please include these in the Reference list (authors, title, publisher (repository name), identifier, year). For more guidance on how to write this section please see:
<http://www.nature.com/authors/policies/data/data-availability-statements-data-citations.pdf>

CODE AVAILABILITY

Please include a “Code Availability” subsection in the Online Methods which details how your custom code is made available. Only in rare cases (where code is not central to the main conclusions of the paper) is the statement “available upon request” allowed (and reasons should be specified).

For more information on our code sharing policy and requirements, please see:
<https://www.nature.com/nature-research/editorial-policies/reporting-standards#availability-of-computer-code>

MATERIALS AVAILABILITY

ORCID

Nature Methods is committed to improving transparency in authorship. As part of our efforts in this direction, we are now requesting that all authors identified as ‘corresponding author’ on published papers create and link their Open Researcher and Contributor Identifier (ORCID) with their account on the Manuscript Tracking System (MTS), prior to acceptance. This applies to primary research papers only. ORCID helps the scientific community achieve unambiguous attribution of all scholarly contributions. You can create and link your ORCID from the home page of the MTS by clicking on ‘Modify my Springer Nature account’. For more information please visit please visit www.springernature.com/orcid.

Please do not hesitate to contact me if you have any questions or would like to discuss these revisions further. We look forward to seeing the revised manuscript and

thank you for the opportunity to consider your work.

Sincerely,
Rita

Rita Strack, Ph.D.
Senior Editor
Nature Methods

Reviewers' Comments:

Reviewer #1:

Remarks to the Author:

Gaining polarization information in fluorescence microscopy allows for observing/quantifying molecular orientation that can be important for many biological studies. There exist many different methods for measuring the polarization in fluorescence microscopy, which are more or less complex and which are extensively discussed in the manuscript. The core topic of the manuscript is now the presentation and application of a new polarization-resolving camera that considerably simplifies the recording of polarization-resolved images. Instead of using complex arrangements of beam-splitters or using polarization modulation in excitation, the new camera records polarization resolved images in situ. Moreover, the authors developed and present extensive freely available software and algorithms for data evaluation, and they developed support of the camera by the widely used image acquisition software MicroManager. This all will make the installation and application of the new camera for polarization-resolved imaging easy for all researchers interested in this kind of microscopy. The authors present two specific applications of the polarization-resolving camera: single-molecule localization and orientation imaging, and diffraction-limited polarization microscopy. I did not find any flaw or missing information in the manuscript, which will be of great interest to anybody interested in polarization-resolved wide-field imaging, and I recommend therefore publication as is.

Reviewer #2:

Remarks to the Author:

In this work, Bruggman et al. presented a new single-molecule dipole orientation imaging technique termed POLCAM, based on the application of a quadratic linear polarization camera. This approach significantly simplifies the conventional multiple-camera or multiple-exposure approach, just like the Bayer filter RGB camera vs. three-CCD RGB imaging or RGB sequential filtering. Likewise, the consequence is a scarification of the effective photon collection, with a shifted pixel position along each polarization angle. The authors carefully evaluated the effect of pixel size, detection position difference, and polarization angle, with proper interpolation. Stokes parameter estimation in the Fourier domain was used to minimize IFOV. PSF fitting using simulated images of fluorescent molecules was employed to improve the angular accuracy of out-of-plane orientation estimation, and an open-source napari

software was developed to achieve real-time polarization display under diffraction-limited conditions.

The overall strengths of this article are:

- λ Considering the inherent errors between polarization camera channels and comparing different methods;
- λ Addressing the scenario of out-of-plane angles in the presence of noise;
- λ Providing a powerful and easy-to-use tool.

To make the technique reach Nature Methods level, the following questions should be well considered:

1. In POLCAM, the pixel is not connected, but separated by 1 pixel for each polarization. This affects the estimation accuracy of the polarized single-molecule, where the weight is coupled with the position.
 - a) The physical size of CMOS camera's pixel is 3.45 μm , whereas the pixel size for a EMCCD camera can be as large as 16 μm . The quantum yield of this camera (CS505MUP, Thorlabs) is only 72%, and the readout noise is much higher than that of the camera traditionally used in single-molecule imaging. I am concerned about its practical application in single-molecule imaging. What's the excitation intensity and whether it is higher than that of traditional polarized single-molecule imaging? How may the noise of the CCD, and this relatively small pixel area contribute to the accuracy of the localization?
 - b) The authors may envision the potential application of sCMOS into POLCAM to further improve the performance of the system.
2. An experimental validation with a 50:50 image splitter, to split the image onto both conventional sCMOS or EMCCD, and POLCAM comparison should be performed, to prove that the position of the single-molecule can be accurately determined in this microscopy system with a discrete quadratic sampling, despite of the polarization angle induced pixels shift.
3. Fig. 1 shows that when circularly polarized light is used for three-dimensional dipole imaging, Gaussian point-like samples will exhibit a ring-shaped distribution, which is very beneficial for polarization analysis. However, can this pattern fitting be applied to super-resolution reconstruction and affect the results of super-resolution reconstruction? Can polarization further promote resolution enhancement or achieve frame reduction?
4. In Fig. 1c, the pixels are corresponding to different polarization, but they were all displayed in grayscale. It is difficult to tell the polarization angle of each pixel. For example, it is suspected that the first -45 degree and the 0 degree polarization camera images were placed wrongly. If the authors can use pseudo-color for the quadratic pixels, it will help the readers to easily tell the angle.
5. The quadratic polarization detection is represented by equations (1) and (2) in Line 185 and references (50) is cited. However, the normalization condition in reference (50) is questionable as it does not respect the detection efficiency equals to 1/2 when the maximum collection angle tends to $\pi/2$. Therefore, the authors need to consider the parameters in the model, especially A, B, and C, as these coefficients can affect the total emission intensity, which is crucial in single-molecule localization and orientation measurements.
6. The resolution evaluated by FRC in Fig. 4 for AF488 and AF647 is 42 nm and 26 nm, respectively. However, from the images shown in Fig. 4c, d, and g, the resolution is much lower than this measurement result. Moreover, it can be seen from Fig. 4 that the noise is very large, so the authors should evaluate the resolution more comprehensively.

7. The authors mentioned using the developed POLCAM-live for fast decisions making and minimizing common sources of error in polarimetry. It is recommended that the authors provide a detailed description of the decision-making process for reducing errors in the excitation polarization state. It is also suggested to add a comparison of the results obtained with different excitation polarization states.
8. In Fig. 5d, the Dipole PSF fitting method shows fluctuations in the accuracy of ϕ during high theta deviations, but the light intensity-only solution method does not show such fluctuations. Please explain the reason for this.
9. How to choose the neighborhood size when calculating netDoLP?
10. In formulas S32 and S38 in SI, both angles are calculated using the intensity in the neighborhood as weights. Is this reasonable? Will weighting by intensity lead to the loss of weak signals with strong polarization? Should DoLP be used as a weight instead?
11. Can polarization imaging itself further improve the spatial resolution through separating two adjacent dipoles with same/different orientations? The authors should do some comparisons and evaluations in this regard.
12. To help the biological users to adapt this technique in their own research, the author wrote a napari plugin, yet it is implemented based on Python and is difficult for biologists to use. If the author can provide an Imagej plugin, I believe it can further increase the applicability of this method.
13. In the results of live cells in Fig. 6, both the signal-to-noise and resolution are much worse than those of fixed cells in 6c. Is this due to motion artifacts? How many images need to be collected to reconstruct a 3D image of a live cell, and what is the total time required?

Minor comments:

1. It is suggested that the format of Fig. 1b be the same as Fig. 2h, as the angle labeled in Fig. 2h is clearer.
2. In formulas 4a-4c, I135 is marked instead of I-45, which is inconsistent with the reference and Fig. 1.
3. In the "Methods" section, "Live-cell imaging of the plasma membrane of Jurkat T cells" is repeated twice.
4. In supplement materials, what is the meaning of the symbol 'n' in Eq. (S3)?
5. 'ADU' appears for the first time in S1.6, but no full name is given.
6. The symbol 'Ttot' in the third line of the first paragraph in S2.3 does not coincide with 'Itot' in Eq. (21).
7. In S3.2.2, what does '... is ...' in the sentence "where ... is ... Equation (S31) was derived by Thompson et al." specifically refer to?
8. There are cross-referencing errors for the figures in lines 333, 335, and 342.

Reviewer #3:

Remarks to the Author:

The manuscript presents a single-molecule orientation localization microscopy method that is faster and simpler to implement than prior techniques. The method is based on the detection of the polarization for every location using a commercial polarization camera.

The technique provides molecular anisotropy information with fast algorithms.

The technique is useful to determine molecular orientation and rotation, which could be of interest in understanding some biological systems.

The key innovation of this report is the use of a commercial polarization camera. The advantage is in speed and simplicity even though it is quite straightforward from the point of view of the hardware. Like prior methods, the manuscript presents vectorial simulations.

- The key disadvantage of this method is the loss of photons at the micro-polarizers of the camera. This leads to SNR and precision penalty with respect to prior techniques.
- The manuscript presents the average degree of linear polarization (avgDoLP) as a proxy for rotational mobility; What is the exact mathematical relation between avgDoLP and the rotational mobility parameter Γ ?
- In the manuscript: "To demonstrate that a polarization camera can also be used for polarized diffraction-limited microscopy"
There is no need to prove this because that is stated in the camera manufacturer specification and the purpose for selling it. There is no reason to assume the images would not be diffraction limited.
- In the manuscript: "We define the optimal pixel size as the largest pixel size that still allows for accurate recovery of the four polarized channels from a single polarization camera image."
Based on this statement, it is not clear what is optimized. It seems there is an "accurate" level accepted (but not optimal) that is not specified.
- In the manuscript: "Operationally, there is no need for polarization channel dependent aberration correction, as all channels are perfectly at the same focal plane and any aberrations will also be identical in all four channels."
This statement is not accurate in general (and thus should be justified in the current system) because aberrations might be different for each polarization (albeit symmetric), especially for high NA imaging, and hence affect the Stokes parameters.
- In the manuscript: "developed a Stokes parameter estimation-based reconstruction algorithm, and a dipole PSF-fitting algorithm."
It should be noted that a dipole does not have a PSF. A dipole has a dipole spread function or Green tensor response.

Author Rebuttal to Initial comments

Rebuttal

Please find attached a point by point rebuttal which includes:

- 1 new experiment
- 2 edited figures in the main manuscript (Fig 1 and Fig 4)
- 1 new column in Table S3 with measured power densities
- 3 new sections in the supplementary information (sections S2, S3 and S7)
- 20 new supplementary figures (S2-S18, S26, S27, S29)

Point-by-point rebuttal

Reviewer #1:

Remarks to the Author: Gaining polarization information in fluorescence microscopy allows for observing/quantifying molecular orientation that can be important for many biological studies. There exist many different methods for measuring the polarization in fluorescence microscopy, which are more or less complex and which are extensively discussed in the manuscript. The core topic of the manuscript is now the presentation and application of a new polarization-resolving camera that considerably simplifies the recording of polarization-resolved images. Instead of using complex arrangements of beam-splitters or using polarization modulation in excitation, the new camera records polarization resolved images in situ. Moreover, the authors developed and present extensive freely available software and algorithms for data evaluation, and they developed support of the camera by the widely used image acquisition software MicroManager. This all will make the installation and application of the new camera for polarization-resolved imaging easy for all researchers interested in this kind of microscopy. The authors present two specific applications of the polarization-resolving camera: single-molecule localization and orientation imaging, and diffraction-limited polarization microscopy. I did not find any flaw or missing information in the manuscript, which will be of great interest to anybody interested in polarization-resolved wide-field imaging, and I recommend therefore publication as is.

We thank the reviewer for their kind support of the paper.

Reviewer #2:

Remarks to the Author: In this work, Bruggman et al. presented a new single-molecule dipole orientation imaging technique termed POLCAM, based on the application of a quadratic linear polarization camera. This approach significantly simplifies the conventional multiple-camera or multiple-exposure approach, just like the Bayer filter RGB camera vs. three-CCD RGB imaging or RGB sequential filtering. Likewise, the consequence is a scarification of the effective photon collection, with a shifted pixel position along each polarization angle. The authors carefully evaluated the effect of pixel size, detection position difference, and polarization angle, with proper interpolation. Stokes parameter estimation in the Fourier domain was used to minimize IFOV. PSF fitting using simulated images of fluorescent molecules was employed to improve the angular accuracy of out-of-plane orientation estimation, and an open-source napari software was developed to achieve real-time polarization display under diffraction-limited conditions.

The overall strengths of this article are:

Considering the inherent errors between polarization camera channels and comparing different methods;

Addressing the scenario of out-of-plane angles in the presence of noise;

Providing a powerful and easy-to-use tool.

We thank the reviewer for their detailed comments on the manuscript and the supplementary material.

To make the technique reach Nature Methods level, the following questions should be well considered:

1) In POLCAM, the pixel is not connected, but separated by 1 pixel for each polarization. This affects the estimation accuracy of the polarized single-molecule, where the weight is coupled with the position.

a) The physical size of CMOS camera's pixel is 3.45 μm , whereas the pixel size for a EMCCD camera can be as large as 16 μm . The quantum yield of this camera (CS505MUP, Thorlabs) is only 72%, and the readout noise is much higher than that of the camera traditionally used in single-molecule imaging. I am concerned about its practical application in single-molecule imaging. What's the excitation intensity and whether it is higher than that of traditional polarized single-molecule imaging?

We have added a column to Supplementary Table S3 with the measured power densities at the sample plane in kW/cm^2 for all experimental data shown in the manuscript. The protocol used for measuring the power density is explained in the figure caption. The power densities used for single-molecule experiments presented in this work are on the order of $0.004 \text{ kW}/\text{cm}^2$ for live cell experiments, to $2.4\text{--}6.1 \text{ kW}/\text{cm}^2$ for dSTORM and PAINT experiments. These are similar to power densities used in single-molecule localization microscopy methods such as dSTORM (Diekmann et al, Nature Methods, 2020, <https://doi.org/10.1038/s41592-020-0918-5>). **Outcome: a new column with measured excitation power densities was added to Supplementary Table S3.**

Figure Ref.	Sample	Dye	Dichroic mirror	Emission filter(s)	Wavelength laser (nm)	Power density (kW/cm ²)	Exposure time (ms)
1a-c	Dye on glass	SYTOX Orange	D03-R532-t1	FF01-58264	520	0.36	100
1e, 6a	Lipid-bilayer bead	Ds-8-ANEPPS	D03-R514-t1	FF01-515LP, FF01-650/200	514	2.02	200
1f	Lipid-bilayer bead	NR4A	D03-R514-t1	FF01-515LP, FF01-650/200	514	6.08	50
1g	Lipid-bilayer bead	Nile red	D03-R514-t1	FF01-515LP, FF01-650/200	514	6.08	50
1h,j	Dye in polymer	Alexa Fluor 647	D03-R405/488/561/635-t1	BLP01-635R	638	-	10-2,000
2	α -syn fibrils	Nile red	D03-R532-t1	BLP01-532R, FF01-650/200	532	2.42	50
3a-c, 3e-g, 4	dSTORM F-actin HeLa	Alexa Fluor 488	D03-R405/488/561/635-t1	BLP01-488R, FF01-58264	488	5.04	30
3b, 3d-f	dSTORM F-actin HeLa	Alexa Fluor 647	D03-R405/488/561/635-t1	BLP01-635R	638	3.51	30
6c	F-actin COST	SIR-actin	D03-R405/488/561/635-t1	BLP01-635R	638	-	100
6d-e	membrane live T cell	NR4A	D03-R514-t1	FF01-515LP, FF01-650/200	514	0.004	200
6f	membrane live T cell	NR4A	D03-R514-t1	FF01-515LP, FF01-650/200	514	0.004	200
S5	Dye on glass	SYTOX Orange	D03-R532-t1	FF01-58264	520	0.36	100
S13a-d	Lipid-bilayer bead	Ds	D03-R514-t1	FF01-515LP, FF01-58264	514	0.81	200
S13e	Lipid-bilayer bead	Nile red	D03-R514-t1	FF01-515LP, FF01-650/200	514	2.02	200
S13f	Lipid-bilayer bead	Ds-8-ANEPPS	D03-R514-t1	FF01-515LP, FF01-650/200	514	2.02	200
S13g-h	Lipid-bilayer bead	NR4A	D03-R514-t1	FF01-515LP, FF01-650/200	514	6.08	50
S14	α -syn fibrils	Nile red	D03-R532-t1	BLP01-532R, FF01-650/200	532	2.42	50
S15	Dye in polymer	Alexa Fluor 647	D03-R405/488/561/635-t1	BLP01-635R	638	-	10-2,000
S21	dSTORM F-actin HeLa	Alexa Fluor 488	D03-R405/488/561/635-t1	BLP01-488R, FF01-58264	488	5.04	30
S22	Lipid-bilayer bead	NR4A	D03-R514-t1	FF01-515LP, FF01-650/200	514	2.02	200
S23-S25	α -syn fibrils	Nile red	D03-R532-t1	BLP01-532R, FF01-650/200	532	2.42	50

Table S5. Imaging parameters. Experimental parameters used for imaging. All dichroic mirrors and emission filters were bought from Semrock. If an entry in the column Power density is labelled '-', it means that no power density was measured during that specific experiment. The power densities were estimated at the end of experiments by removing the sample and measuring the total power of the excitation beam exiting the objective using a power meter. This value was divided by the area of the illuminated area, which was estimated from the illumination profile visible in the images themselves (which was possible as the image of the illuminated area was smaller than the camera sensor).

How may the noise of the CCD, and this relatively small pixel area contribute to the accuracy of the localization?

The virtual pixel size used in this work (57.5 nm and 69 nm) is smaller than what is typically used in SMLM (100 nm). This means that the signal photons will be spread over more pixels, decreasing the signal-to-noise ratio. This will to some extent negatively influence the localisation precision, but doesn't significantly influence the localisation accuracy¹. In our analysis algorithms, we correct for pixel-dependent camera parameters using experimentally determined camera offset and gain maps.

To demonstrate that we can accurately super-resolve structures experimentally, we have included a new figure in the supplementary material (Figure S29) that shows a reconstruction of DNA origami recorded on our polarisation camera setup.

Outcome: new experimental data and supplementary figure (Fig S29)

Fig. S29. DNA-PAINT of DNA-origami nanorulers with 80 nm spacing. Reconstruction of DNA-origami nanorulers (GATTA-PAINT 80R, Gattaquant2) that were imaged using the polarization camera (20,000 frames, 200 ms exposure time, 4.8 kW/cm² in TIRF). The nanorulers have three docking sites with a 80 nm spacing between docking sites. The imager strands are labeled with ATTO 655. Sample drift was corrected using RGC (15).

¹ Thompson *et al.*, Precise nanometer localization analysis for individual fluorescent probes. Biophysical Journal, 82(5):2775–2783, 2002

- b) The authors may envision the potential application of sCMOS into POLCAM to further improve the performance of the system.

We agree with the reviewer. Currently available polarisation cameras are marketed primarily to industry for applications that are not photon-limited. We hope that a similar push towards lower read noise and increased quantum efficiency will occur for polarisation cameras, as it did for conventional cameras. In fact, we have actively reached out to companies to show our interest in purchasing/co-developing polarisation cameras with, e.g. active cooling, on board correction for column noise, but the timescale for these developments is slow. Hopefully our work might be able to play a part in creating an incentive for companies to develop more advanced polarisation cameras.

- 2) An experimental validation with a 50:50 image splitter, to split the image onto both conventional sCMOS or EMCCD, and POLCAM comparison should be performed, to prove that the position of the single-molecule can be accurately determined in this microscopy system with a discrete quadratic sampling, despite of the polarization angle induced pixels shift.

We have address this point in two ways:

1. Please see the DNA origami experiment (above) that shows that there are no obvious systematic localisation errors at the nanoscale.
2. We performed the proposed experiment *in silico* and summarised the results in a new supplementary figure (Fig S26). We opted to do the suggested comparison using simulations as it makes data interpretation more straightforward. We used the following approach: images of single immobilised fluorescent molecules were simulated at a range of 3D orientations (sampling a hemisphere in steps of 7.5 degrees in phi and theta). For each orientation, a polarisation camera image and a 'regular camera' image was simulated. All images were then localised using least-squares fitting of a rotated asymmetric Gaussian. In the case of the polarisation camera images, the localisation was performed on the S0 image.

As expected, both the polarisation camera and regular camera images result in a slightly biased localisation as a result of the immobilised dipole. Importantly, the bias is the same (less than 1 nm difference at most) between the two cameras. Therefore we conclude that the use of a polarisation camera does not result in significant increase in localisation bias compared to a regular camera.

Outcome: new simulations and supplementary figure (S26), reproduced below

Fig. S26. Comparison of localization bias due to a dipole emitter for detection with a polarization camera and regular camera. The localization bias due to a dipole emitter was compared between a polarization camera and a regular camera. Images of immobilized, single dipole emitters were simulated at different 3D orientations in a hemisphere ($\mu_z \geq 0$), and localized using least-squares fitting of a rotated asymmetric Gaussian. **i)** The localization bias in the x-direction, for a polarization camera. **ii)** The localization bias in the y-direction, for a polarization camera. **iii)** The localization bias in the x-direction, for a regular camera. **iv)** The localization bias in the y-direction, for a regular camera. **v)** The difference between the data in plots (ii) and (i). **vi)** The difference between the data in plots (iv) and (iii).

- 3) Fig. 1 shows that when circularly polarized light is used for three-dimensional dipole imaging, Gaussian point-like samples will exhibit a ring-shaped distribution, which is very beneficial for polarization analysis. However, can this pattern fitting be applied to super-resolution reconstruction and affect the results of super-resolution reconstruction?

In the manuscript we present two super-resolution reconstruction algorithms; 1) an algorithm that only considers polarisation, and 2) an algorithm that considers both polarisation and shape, based on the previously published algorithm RoSE-O². A comparison of the algorithms is presented in figure 5 and Supplementary Figures S33-S37. The result of this comparison is that including the shape indeed increases position and orientation accuracy, but at the expense of processing time. We have adapted the text under section ‘Improving accuracy by considering the DSF shape’ to make this more clear:

“From equation (2), it is clear that the estimation of the polar angle θ will become biased in the presence of rotational mobility, as netDoLP will decrease with increasing rotational mobility. Additionally, equation (2) becomes biased in the presence of noise, and has a dependency on the refractive index of the sample medium. As a result, unbiased estimation of θ using Eq. (2) can only be performed using the previously discussed algorithm when molecules are perfectly immobilized and the signal-to-noise ratio is high. We will now refer to this algorithm as the intensity-only algorithm. We demonstrate that this limitation can be overcome by additionally taking the shape of the DSF into account. To this end, we adapted the previously published DSF-fitting algorithm RoSE-O (66) for use with a polarization

² Mazidi et al., Dense super-resolution imaging of molecular orientation via joint sparse basis deconvolution and spatial pooling. IEEE 16th International Symposium on Biomedical Imaging (ISBI 2019), p325–329, 2019

camera. This algorithm fits the shape of the image of a single emitter in all four polarized channels to estimate the orientation and rotational mobility of the emitter.”

Outcome: Modified text in manuscript

Can polarization further promote resolution enhancement or achieve frame reduction?

Knowing the orientation of single emitters allows for reducing localisation bias due to dipole effects, and one could also imagine that it may be possible to image at higher emitter densities and still resolve partially overlapping emitters due to differences in polarisation. That being said, these topics are outside the scope of the current manuscript, but we certainly agree with the reviewer this is an exciting idea and will be explored further in future work.

- 4) In Fig. 1c, the pixels are corresponding to different polarization, but they were all displayed in grayscale. It is difficult to tell the polarization angle of each pixel. For example, it is suspected that the first -45 degree and the 0 degree polarization camera images were placed wrongly. If the authors can use pseudo-color for the quadratic pixels, it will help the readers to easily tell the angle.

We have modified figure 1 in the main manuscript and added a new column of simulated images to panel 1c that show what the polarisation camera images look like when the pixels are rearranged into the four channels using only pixel rearrangement and no processing (see figure below). This additional visualisation should make it easy for a reader to interpret the pixel values, in a similar way to the suggested colourcoding. We have also included a new supplementary figure (Fig S10) with a colour-coded rendering as suggested by the reviewer.

We also thank the reviewer for pointing out that the simulated image in the first row was incorrectly labelled. We have corrected this in the updated version of the figure.

Outcome: modified figure (Fig 1c) and new supplementary figure (Fig S10), both reproduced below

Fig. S10. Simulated polarization camera images for different dipole orientations: Polarization. Simulated polarisation camera images at the image plane for an immobilized single fluorescent molecule at different 3D orientations, visualized using a polarization colormap that combines AoLP, DoLP, and S_0 in HSV colorspace. Pixels with a micropolarizer oriented at 0° , 45° , 90° and -45° relative to the x-axis will respectively appear cyan, purple, red, and green. The simulation parameters used to generate this figure are described in section S2. The molecules are positioned on a planar refractive index interface between glass ($n=1.158$) and water ($n=1.33$).

- 5) The quadratic polarization detection is represented by equations (1) and (2) in Line 185 and references (50) is cited. However, the normalization condition in reference (50) is questionable as it does not respect the detection efficiency equals to 1/2 when the maximum collection angle tends to \$\pi/2\$. Therefore, the authors need to consider the parameters in the model, especially A, B, and C, as these coefficients can affect the total emission intensity, which is crucial in single-molecule localization and orientation measurements.

To avoid any confusion and for full transparency, we have included the complete derivation of all expressions used in our work in the **Supplementary Material in Section S4**. We note that we do not use equations (1) and (2) to estimate photon numbers; the photon numbers are extracted from the PSF fitting step in the reconstruction algorithm.

Outcome: new section in Supplementary Information (Section S4)

- 6) The resolution evaluated by FRC in Fig. 4 for AF488 and AF647 is 42 nm and 26 nm, respectively. However, from the images shown in Fig. 4c, d, and g, the resolution is much lower than this measurement result. Moreover, it can be seen from Fig. 4 that the noise is very large, so the authors should evaluate the resolution more comprehensively.

We thank the reviewer for pointing this out. We made a normalisation error when generating the FRC curves. This has now been fixed and figure 4f has been regenerated. The calculated FRC resolution of the AF488 and AF647 datasets are now respectively 70 nm and 55 nm.

Outcome: FRC analysis redone and figures and text updated accordingly, reproduced below

- 7) The authors mentioned using the developed POLCAM-live for fast decisions making and minimizing common sources of error in polarimetry. It is recommended that the authors provide a detailed description of the decision-making process for reducing errors in the excitation polarization state. It is also suggested to add a comparison of the results obtained with different excitation polarization states.

We have elaborated on this topic in the main text and added a new section in the supplementary material (Section S7).

Outcome: modified text and supplementary material (Section S7).

- 8) In Fig. 5d, the Dipole PSF fitting method shows fluctuations in the accuracy of \$\phi\$ during high theta deviations, but the light intensity-only solution method does not show such fluctuations. Please explain the reason for this.

These fluctuations are due to the finite number of repeats used in the simulations, the appearance of which is amplified by the fact that both axes are in a log-scale.

- 9) How to choose the neighborhood size when calculating netDoLP?

The choice of neighborhood size for netDoLP was based on simulations. Briefly, we estimated netDoLP from simulated images of a single emitter (including shot noise and camera detection noise matching our camera) as a function of neighborhood size. We performed these simulations for different 3D orientations and rotational mobilities. As expected, a neighborhood size that just fits the image of a single emitter when it is oriented out-of-plane is the optimal neighborhood size. The simulation code used in this work is available on github.

Outcome: A sentence was added to the relevant section of the supplementary material to explain neighbourhood size choice.

“For netDoLP estimation, a value of $m = 15$ was used, motivated by simulations. For the specific optical setup used in this work, this corresponds to the area on the detector that just about contains the image of a single emitter with a dipole moment oriented parallel to the optical axis (i.e., the orientation with the largest spatial footprint).”

- 10) In formulas S32 and S38 in SI, both angles are calculated using the intensity in the neighborhood as weights. Is this reasonable? Will weighting by intensity lead to the loss of weak signals with strong polarization? Should DoLP be used as a weight instead?

The intensity weighting (S0) was implemented based on the analysis of simulated images. We compared weighting by DoLP, S0 and combinations of DoLP and S0. Weighting by S0 resulted in the highest accuracy, and rationalised by the fact that high

frequency noise (such as the background area around single emitters) will result in a high DoLP, this is avoided when weighted by intensity. This approach is common in analogous spectroscopy methods for instance Fluorescence lifetime imaging where this has been common in commercial systems for decades³.

- 11) Can polarization imaging itself further improve the spatial resolution through separating two adjacent dipoles with same/different orientations? The authors should do some comparisons and evaluations in this regard.

We agree with the reviewer that this is an exciting area that should be explored further in the future. In our current work we made sure that all samples were sufficiently sparse that overlapping emitters were extremely rare, as implementing a multi-emitter fitting is outside the scope of this paper.

- 12) To help the biological users to adapt this technique in their own research, the author wrote a napari plugin, yet it is implemented based on Python and is difficult for biologists to use. If the author can provide an Imagej plugin, I believe it can further increase the applicability of this method.

Napari can be installed as a standalone software and used without any knowledge of python. Aside from the napari plugin, we also developed standalone MATLAB software with a single-click installation and very intuitive GUI, which also does not require the user to have any programming experience. We would like to make the comparison here with the software SMAP⁴, which despite being developed in MATLAB, also has found widespread application.

That being said, we agree with the reviewer that an ImageJ plugin would further increase the applicability of the method. Given sufficient interest, we would happily develop a polarisation camera image processing plugin for ImageJ in the future, however we do not want this to affect the timescale of getting the manuscript to the community.

- 13) In the results of live cells in Fig. 6, both the signal-to-noise and resolution are much worse than those of fixed cells in 6c. Is this due to motion artifacts? How many images need to be collected to reconstruct a 3D image of a live cell, and what is the total time required?

The difference in signal-to-noise ratio between the fixed cell and live cell datasets is to a large extent due to the different labelling approaches that were used: SiR-actin for fixed cells, versus the membrane stain NR4A for live cells. The two datasets were also rendered using different software; the fixed cell 2D data was rendered in MATLAB, and the 3D live-cell data was rendered in napari.

We do think that some motion blur is occurring in the live-cell data, as our imaging speed was limited by the speed of the z-scanning to ~1.6 slices per second. To acquire a full cell, a z-stack of around 80 slices was acquired (with 200 nm separation between slices) at ~1.6 slices per second (the maximum speed allowed by our current setup), resulting in **49.6 seconds/cell volume**. In principle, with a faster z-scanning setup, it should be possible to image much faster, but our optical setup was not designed for fast

³<https://www.picoquant.com/products/category/fluorescence-microscopes/microtime-200-time-resolved-confocal-fluorescence-microscope-with-unique-single-molecule-sensitivity>

⁴ Jonas Ries, SMAP: a modular super-resolution microscopy analysis platform for SMLM data. Nature Methods, 17:870–872, 2020.

3D imaging, this is not a limitation of POLCAM, but of the current instrumental implementation

Minor comments:

- 1) It is suggested that the format of Fig. 1b be the same as Fig. 2h, as the angle labeled in Fig. 2h is clearer. While we strongly considered this helpful idea, external review from microscopy colleagues suggested incorporating this change might unintentionally introduce complexity to the figure. With the aim of maintaining clarity and simplicity, we have kept the figure as originally presented.
- 2) In formulas 4a-4c, I135 is marked instead of I-45, which is inconsistent with the reference and Fig. 1.
Changed.
- 3) In the "Methods" section, "Live-cell imaging of the plasma membrane of Jurkat T cells" is repeated twice.
Copy was removed.
- 4) In supplement materials, what is the meaning of the symbol 'n' in Eq. (S3)?
Definition was added.
- 5) 'ADU' appears for the first time in S1.6, but no full name is given.
Full name added.
- 6) The symbol 'Ttot' in the third line of the first paragraph in S2.3 does not coincide with 'tot' in Eq. (21).
Changed.
- 7) In S3.2.2, what does '... is ...' in the sentence "where ... is ... Equation (S31) was derived by Thompson et al." specifically refer to?
Definitions added.
- 8) There are cross-referencing errors for the figures in lines 333, 335, and 342.
Corrected.

Reviewer #3:

Remarks to the Author: The manuscript presents a single-molecule orientation localization microscopy method that is faster and simpler to implement than prior techniques. The method is based on the detection of the polarization for every location using a commercial polarization camera.

The technique provides molecular anisotropy information with fast algorithms.

The technique is useful to determine molecular orientation and rotation, which could be of interest in understanding some biological systems.

The key innovation of this report is the use of a commercial polarization camera. The advantage is in speed and simplicity even though it is quite straightforward from the point of view of the hardware. Like prior methods, the manuscript presents vectorial simulations.

The key disadvantage of this method is the loss of photons at the micro-polarizers of the camera. This leads to SNR and precision penalty with respect to prior techniques.

- 1) The manuscript presents the average degree of linear polarization (avgDoLP) as a proxy for rotational mobility. What is the exact mathematical relation between avgDoLP and the rotational mobility parameter Gamma?

We used avgDoLP as a proxy for rotational mobility as the exact mathematical relation between avgDoLP and gamma is highly complex and a derivation is outside the scope of this work. Therefore, we mention in the manuscript that avgDoLP should only be used qualitatively (e.g., to compare between two labeling methods as done in figure 4), and that if an accurate estimate of gamma is required, the user should opt for our dipole-spread function fitting-based algorithm, rather than the intensity-only algorithm. We have added new figures to the supplementary material (Figures S9, S13-S18) and edited the main manuscript to make this more clear to the reader:

"The exact mathematical relation between avgDoLP and γ is complex as avgDoLP is also influenced by the signal-to-noise ratio. Nevertheless, taking this dependence into consideration, avgDoLP can still be used to qualitatively assess rotational mobility (Supplementary Figures S13 and S17)."

Outcome: modified text and new supplementary figures (Fig. S9, S13-S18)

- 2) In the manuscript: "To demonstrate that a polarization camera can also be used for polarized diffraction-limited microscopy". There is no need to prove this because that is stated in the camera manufacturer specification and the purpose for selling it. There is no reason to assume the images would not be diffraction limited.

We have rephrased this sentence in the manuscript to "To demonstrate that a polarization camera can also be used for conventional polarized detection microscopy". Our intention here was to demonstrate that a polarisation camera can also be used in combination with conventional, non-single-molecule fluorescence microscopy, which might not be obvious to readers that are unfamiliar with polarimetry.

Outcome: modified text

- 3) In the manuscript: "We define the optimal pixel size as the largest pixel size that still allows for accurate recovery of the four polarized channels from a single polarization

20

camera image." Based on this statement, it is not clear what is optimized. It seems there is an "accurate" level accepted (but not optimal) that is not specified.

We have added a description of our criterion to the text:

"We define the optimal pixel size as the largest pixel size that still allows for accurate recovery of the four polarized channels from a single polarization camera image. To assess whether accurate recovery is possible, we used an approach described by Tyo *et al.* (45) that checks for overlap between the contributions of different Stokes parameters in the Fourier transform of the unprocessed polarization camera image (Supplementary Note S5.1). *If the contributions don't overlap, the recovery is assumed to be accurate.*"

Outcome: modified text

- 4) In the manuscript: "Operationally, there is no need for polarization channel dependent aberration correction, as all channels are perfectly at the same focal plane and any aberrations will also be identical in all four channels." This statement is not accurate in general (and thus should be justified in the current system) because aberrations might be different for each polarization (albeit symmetric), especially for high NA imaging, and hence affect the Stokes parameters.

The statement has been reworded in the manuscript to more clearly portray its intended meaning:

"In conventional four-channel polarized detection where beam splitters and polarization optics are used to separate the fluorescence into four channels, the photons that reach each channel will have traveled along different optical paths. As a result, it is more likely that channel-dependent aberrations will occur due to imperfections in the alignment and optical elements. In contrast, a benefit of the use of a polarization camera is that all of the detected fluorescence travels along the same optical path, simplifying the use of a DSF fitting algorithm."

Outcome: modified text

- 5) In the manuscript: "developed a Stokes parameter estimation-based reconstruction algorithm, and a dipole PSF-fitting algorithm." It should be noted that a dipole does not have a PSF. A dipole has a dipole spread function or Green tensor response.

All mentions of "dipole PSF" in the manuscript, supplementary material and figures were replaced by "dipole spread function" or the abbreviation "DSF".

Outcome: modified text

Decision Letter, first revision:

Date: 8th Jan 24 16:23:32
Last Sent: 8th Jan 24 16:23:32
Triggered By: Rita Strack
From: rita.strack@us.nature.com
To: sl591@cam.ac.uk
CC: methods@us.nature.com
Subject: Decision on Nature Methods submission NMETH-A51455A
Message: 8th Jan 2024

Dear Steven,

Thank you for your letter detailing how you would respond to the reviewer concerns regarding your Article, "POLCAM: Instant molecular orientation microscopy for the life sciences". We have decided to invite you to revise your manuscript as you have outlined, before we reach a final decision on publication.

[REDACTED]

We hope to receive your revised paper within four weeks. If you cannot send it within this time, please let us know. In this event, we will still be happy to reconsider your paper at a later date so long as nothing similar has been accepted for publication at Nature Methods or published elsewhere.

OPEN SCIENCE REQUIREMENTS

REPORTING SUMMARY AND EDITORIAL POLICY CHECKLISTS

DATA AVAILABILITY

Please include a "Data availability" subsection in the Online Methods. This section should inform readers about the availability of the data used to support the conclusions of your study, including accession codes to public repositories, references to source data that may be published alongside the paper, unique identifiers such as URLs to data repository entries, or data set DOIs, and any other statement about data availability. At a minimum, you should include the following statement: "The data that support the findings of this study are available from the corresponding author upon request", describing which data is available upon request and mentioning any restrictions on availability. If DOIs are provided, please include these in the Reference list (authors, title, publisher (repository name), identifier, year). For more guidance on how to write this section please see:

<http://www.nature.com/authors/policies/data/data-availability-statements-data-citations.pdf>

CODE AVAILABILITY

Please include a "Code Availability" subsection in the Online Methods which details how your custom code is made available. Only in rare cases (where code is not central to the main conclusions of the paper) is the statement "available upon request" allowed (and reasons should be specified).

For more information on our code sharing policy and requirements, please see: <https://www.nature.com/nature-research/editorial-policies/reporting-standards#availability-of-computer-code>

MATERIALS AVAILABILITY

ORCID

Nature Methods is committed to improving transparency in authorship. As part of our efforts in this direction, we are now requesting that all authors identified as 'corresponding author' on published papers create and link their Open Researcher and Contributor Identifier (ORCID) with their account on the Manuscript Tracking System (MTS), prior to acceptance. This applies to primary research papers only. ORCID helps the scientific community achieve unambiguous attribution of all scholarly contributions. You can create and link your ORCID from the home page of the MTS by clicking on 'Modify my Springer Nature account'. For more information please visit please visit www.springernature.com/orcid.

Sincerely,
Rita

Rita Strack, Ph.D.
Senior Editor

Nature Methods

Reviewers' Comments:

Reviewer #2:

Remarks to the Author:

In this revision, the authors have addressed all the previous questions with additional DNA origami experiment and indepth simulations. Overall, I am convinced that POLCAM will be a powerful tool for single-molecule polarization microscopy. With further analyze the polarization of the DNA origami, it may be a good case to demonstrate the full potential of POLCAM. The manuscript can be accepted after some minor revisions:

1. Please analyze and present the orientation results of DNA origami in Fig S29, referring to the orientation results in Guan et al. *Light: Science & Applications* (2022) 11:4 (Fig.2). If the polarization is constant, the authors may consider placing it in the maintext.
2. It is recommended to consider these relevant articles for further discussion and extension of this idea, as they and POLCAM belong to active polarization modulation/passive polarization detection, respectively:
[1] Hafi, N., et al. "Fluorescence nanoscopy by polarization modulation and polarization angle narrowing." *Nature methods* 11.5 (2014): 579-584.
[2] Zhanghao, K., et al. "Super-resolution dipole orientation mapping via polarization demodulation." *Light: Science & Applications* 5.10 (2016): e16166.

Reviewer #3:

Remarks to the Author:

The reviewers have addressed some but not all of the concerns raised by the reviewers:

1. Reviewer 2 requested: "An experimental validation with a 50:50 image splitter, to split the image onto both conventional sCMOS or EMCCD, and POLCAM comparison should be performed"... "To make the technique reach Nature Methods level" in line with reviewer 3 comment that "The key disadvantage of this method is the loss of photons at the micro-polarizers of the camera."

The authors present instead a simulation and refer to the DNA origami experiment. While instructive, the simulation might not take into account all the non-ideal features of the experiment and camera.

2. To the comment that the average degree of linear polarization (avgDoLP) as a proxy for rotational mobility without mathematical rigor, the authors responded the derivation is "outside the scope of this work". It is hard to justify the use of a metric in qualitative form if there is no clear understanding of the relation to the magnitude being described.

Is the relation monotonic, linear, non-linear, valid in some range or under some conditions? Following Reviewer 2 comment, this does not "make the technique reach Nature Methods level".

3. Reviewer 3 stated: In the manuscript: "To demonstrate that a polarization camera can also be used for polarized diffraction-limited microscopy". There is no need to prove this because that is stated in the camera manufacturer specification and the purpose for selling it. There is no reason to assume the images would not be diffraction limited.

The authors modified the text but the abstract still reads: "To demonstrate that POLCAM also allows diffraction-limited imaging". The authors should provide a straight explanation that the camera is also good for polarization microscopy and describe the example experiment. I would leave this to the supplementary documents since the novelty is minor compared to the rest of the manuscript.

This is a quality work that merits publication but still needs some work to be published in Nature Methods.

Author Rebuttal, first revision:

Reviewers' Comments

Reviewer #2:

In this revision, the authors have addressed all the previous questions with additional DNA origami experiment and indepth simulations. Overall, I am convinced that POLCAM will be a powerful tool for single-molecule polarization microscopy. With further analyze the polarization of the DNA origami, it may be a good case to demonstrate the full potential of POLCAM. The manuscript can be accepted after some minor revisions:

1. Please analyze and present the orientation results of DNA origami in Fig S29, referring to the orientation results in Guan et al. Light: Science & Applications (2022) 11:4 (Fig.2). If the polarization is constant, the authors may consider placing it in the maintext.

We performed exactly as requested and have generated a new SI figure summarising a polarisation analysis of our DNA-origami dataset:

Panels a and b show the distribution of detected photons and the avgDoLP of the different localisations. The avgDoLP is low compared to other datasets presented in this work (e.g., Fig. 3g and 4e), suggesting that the dyes on the imager strands are rotationally free relative to the DNA origami structure.

Panel c, d and e: When we plot the number of detected photons, avgDoLP and the in-plane angle against each other, we see distributions consistent with rotationally free emitters:

In **panel f**, we plot the reconstructed DNA-origami data in two colours. All localisations with an avgDoLP larger than 0.4 are magenta, and all with an avgDoLP smaller than 0.4 are cyan. It is clear that nearly all localisations

associated with origami are cyan, magenta localisations are likely probably free imager strand binding non-specifically to the surface.

In **panel g and h**, we plot the same data as in panel f, but with a 2D colourmap that encodes both the avgDoLP (from white to coloured) and the in-plane angle phi (rainbow). The map can be interpreted as follows: white dots have low avgDoLP, and coloured dots have high avgDoLP. If localisations in a cluster have the same colour, they are oriented in the same direction. Many non-specific binding events seem to have high avgDoLP and be oriented in the same direction, supporting the idea that these might be free dye binding to the surface.

These observations are in contrast with the data shown in Guan et al. *Light: Science & Applications* (2022) 11:4 (Fig.2), which shows an example of an origami where the dyes in the individual binding domains seem to have similar orientations and less rotational freedom:

This might be due to a difference in sample preparation, imaging speed, or the method. In future work, our hope is that POLCAM will be adopted to study these difference in a reproducible way across labs.

2. It is recommended to consider these relevant articles for further discussion and extension of this idea, as they and POLCAM belong to active polarization modulation/passive polarization detection, respectively:

- Hafi, N., et al. "Fluorescence nanoscopy by polarization modulation and polarization angle narrowing." *Nature methods* 11.5 (2014): 579-584.
- Zhanghao, K., et al. "Super-resolution dipole orientation mapping via polarization demodulation." *Light: Science & Applications* 5.10 (2016): e16166.

We have discussed and included these three references.

Reviewer #3:

The reviewers have addressed some but not all of the concerns raised by the reviewers:

1. Reviewer 2 requested: "An experimental validation with a 50:50 image splitter, to split the image onto both conventional sCMOS or EMCCD, and POLCAM comparison should be performed"... "To make the technique reach *Nature Methods* level" in line with reviewer 3 comment that "The key disadvantage of this method is the loss of photons at the micro-polarizers of the camera."

We performed the proposed experiment and imaged fluorescent beads and single molecules immobilised in polymer simultaneously on the polarisation camera and a state-of-the-art sCMOS camera (Prime 95B, Photometrics) using a 50:50 beam splitter setup. The results are summarised in a new SI figure (panels shown below) and show that we can detect fluorescent beads (panels b, c and d) and single-molecules (panel e) simultaneously on a state-of-the-art sCMOS camera and our polarisation camera.

Panel a shows the setup that was used:

The relay lenses were chosen such that the virtual pixel size on both cameras is like what would be used in a regular experiment: 110 nm for the sCMOS camera, and 69 nm for the polarisation camera.

Panel b shows that we can register the images from both cameras onto each other using a similarity transform (only translation, rotation and global scaling):

Panel c shows examples of the repeated localisation of single beads on both cameras for various signal levels. Each bead was localised 100 times on each camera. The localisations of some representative beads are shown as scatter plots. Above each scatter plot, the mean number of photons detected per frame and the standard deviation on the localisation coordinates are provided. The six examples shown are for the same 6 beads on each camera. As expected, the polarisation camera detects about 50% of the photons compared to the Prime 95B, as the polarisers absorb light. Secondly, the localisation precision gets better (standard deviation smaller) when more photons are detected.

Panel d shows that the polarisation camera detects half as many photons as the Prime 95B. Each dot on the graph represents the average number of photons detected per frame from one single bead, on both cameras. The average is calculated from 100 repeats. Photon numbers are calculated by taking into account the camera gain and wavelength-dependent quantum efficiency. The grey line is the function $y(x) = x/2$. We do indeed expect the data to follow this trend, as the emission of beads will be approximately randomly polarised (a bead is a collection of randomly oriented dyes), and according to Malus' Law, randomly polarised light will be attenuated by 50% after passing through a linear polariser:

Panel e shows that we can detect a single molecule on both cameras simultaneously. Single Cy5 molecules embedded and immobilised in polymer (PMMA) were imaged with an exposure time of 200 ms. The example shown is in the top 5% brightest emitters in the collected dataset. The number of detected photons is clearly correlated between the two cameras; we observe one blinking event, followed by permanent photobleaching in a single step, characteristic of single fluorescent molecules:

2. The authors present instead a simulation and refer to the DNA origami experiment. While instructive, the simulation might not take into account all the non-ideal features of the experiment and camera.

To the comment that the average degree of linear polarization (avgDoLP) as a proxy for rotational mobility without mathematical rigor, the authors responded the derivation is "outside the scope of this work". It is hard to justify the use of a metric in qualitative form if there is no clear understanding of the relation to the magnitude being described.

Is the relation monotonic, linear, non-linear, valid in some range or under some conditions? Following Reviewer 2 comment, this does not "make the technique reach Nature Methods level".

We numerically calculated the relation between avgDoLP and gamma, and summarised the results in a new SI figure:

Although the exact relationship between gamma and avgDoLP depends on a number of variables (such as out-of-plane angle, SNR and the refractive index of the sample medium), the relationship is monotonic and allows qualitative comparisons between samples as asked by the reviewer. We reference to this new figure in the main manuscript and in the Supplementary material:

“The relationship between avgDoLP and rotational mobility is numerically explored in Fig. S27, showing that it is monotonic under all conditions and can therefore be used qualitatively, but with care.”

- Reviewer 3 stated: In the manuscript: “To demonstrate that a polarization camera can also be used for polarized diffraction-limited microscopy”. There is no need to prove this because that is stated in the camera manufacturer specification and the purpose for selling it. There is no reason to assume the images would not be diffraction limited.

The authors modified the text but the abstract still reads: “To demonstrate that POLCAM also allows diffraction-limited imaging”. The authors should provide a

straight explanation that the camera is also good for polarization microscopy and describe the example experiment. I would leave this to the supplementary documents since the novelty is minor compared to the rest of the manuscript.

We thank the reviewer for pointing this out. We have now also removed this sentence from the abstract.

This is a quality work that merits publication but still needs some work to be published in Nature Methods.

Decision Letter, second revision:

Date: 9th Feb 24 10:31:45
Last Sent: 9th Feb 24 10:31:45
Triggered By: Rita Strack
From: rita.strack@us.nature.com
To: sl591@cam.ac.uk
CC: methods@us.nature.com
Subject: AIP Decision on Manuscript NMETH-A51455B
Message: Our ref: NMETH-A51455B

9th Feb 2024

Dear Steven,

Thank you for submitting your revised manuscript "POLCAM: Instant molecular orientation microscopy for the life sciences" (NMETH-A51455B). It has now been seen by the original referees and their comments are below. The reviewers find that the paper has improved in revision, and therefore we'll be happy in principle to publish it in Nature Methods, pending minor revisions to comply with our editorial and formatting guidelines.

TRANSPARENT PEER REVIEW

Please note: we allow redactions to authors' rebuttal and reviewer comments in the interest of confidentiality. If you are concerned about the release of confidential data, please let us know specifically what information you would like to have removed. Please note that we cannot incorporate redactions for any other reasons. Reviewer names will be published in the peer review files if the reviewer signed the comments to authors, or if reviewers explicitly agree to release their name. For more information, please refer to our FAQ page.

ORCID

Sincerely,
Rita

Rita Strack, Ph.D.
Senior Editor
Nature Methods

Reviewer #2 (Remarks to the Author):

The authors have addressed all the previous concerns with additional experiments on the fluorescence orientation of DNA origami, and the side-by-side comparison of POLCAM and sCMOS detector. I am satisfied with the results, and convinced that the paper is acceptable for Nature Methods in its current form.

Final Decision Letter:

Dear Steve,

Please let me begin by apologizing for the delays in processing your paper.

I am pleased to inform you that your Article, "POLCAM: Instant molecular orientation microscopy for the life sciences", has now been accepted for publication in Nature Methods. The received and accepted dates will be Feb 3, 2023 and July 17, 2024. This note is intended to let you know what to expect from us over the next month or so, and to let you know where to address any further questions.

Over the next few weeks, your paper will be copyedited to ensure that it conforms to Nature Methods style. Once your paper is typeset, you will receive an email with a link to choose the appropriate publishing options for your paper and our Author Services team will be in touch regarding any additional information that may be required. It is extremely important that you let us know now whether you will be difficult to contact over the next month. If this is the case, we ask that you send us the contact information (email, phone and fax) of someone who will be able to check the proofs and deal with any last-minute problems.

Please note that *Nature Methods* is a Transformative Journal (TJ). Authors may publish their research with us through the traditional subscription access route or make their paper immediately open access through payment of an article-processing charge (APC). Authors will not be required to make a final decision about access to their article until it has been accepted. Find out more about Transformative Journals

If you are active on Twitter/X, please e-mail me your and your coauthors' handles so that we may tag you when the paper is published.

Best regards,
Rita

Rita Strack, Ph.D.
Senior Editor
Nature Methods